# VIDEOVERITAS: AI-Generated Video Detection via Perception Pretext Reinforcement Learning

Hao Tan [1 2 3]  Jun Lan [† 3]  Senyuan Shi [2]  Zichang Tan [4]  Zijian Yu [3]  Huijia Zhu [3]  Weiqiang Wang [3]
Jun Wan [1 2 5]  Zhen Lei [1 2 5]

 Project: https://github.com/EricTan7/VideoVeritas

## Abstract

The growing capability of video generation poses escalating security risks, making reliable detection increasingly essential. In this paper, we introduce VIDEOVERITAS, a framework that integrates fine-grained perception and fact-based reasoning. We observe that while current multimodal large language models (MLLMs) exhibit strong reasoning capacity, their granular perception ability remains limited. To mitigate this, we introduce *Joint Preference Alignment* and *Perception Pretext Reinforcement Learning (PPRL)*. Specifically, rather than directly optimizing for detection task, we adopt general spatiotemporal grounding and self-supervised object counting in the RL stage, enhancing detection performance with simple *perception pretext tasks*. To facilitate robust evaluation, we further introduce **MintVid**, a light yet high-quality dataset containing 3K videos from 9 state-of-the-art generators, along with a real-world collected subset that has factual errors in content. Experimental results demonstrate that existing methods tend to bias towards either *superficial* reasoning or *mechanical* analysis, while VIDEOVERITAS achieves more balanced performance across diverse benchmarks.

## 1. Introduction

With the rapid advancement of video generation, our digital lives have been immensely enriched. AI-generated videos

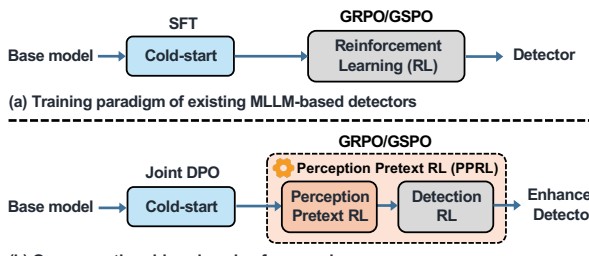

*Figure 1.* **Comparison with previous training pipeline.** (a) Existing MLLM-based detectors typically adopt supervised fine-tuning (SFT) or reinforcement learning (RL) on detection task. (b) Our framework adopts Joint DPO for cold-start, and further enhances the detection capacity by introducing simple perception pretext tasks in the RL stage.

have become pervasive on short-video platforms such as Tik-Tok, which, while providing unprecedented entertainment, raise significant concerns regarding security. AI-generated video detection, which aims at determining the authenticity of a video, has therefore emerged as a critical topic.

In this landscape, various methods (Chen et al., 2024; Ma et al., 2025; Zheng et al., 2025b; Internò et al., 2025; Zhang et al., 2025b; Corvi et al., 2025) and datasets (Chen et al., 2024; Ni et al., 2025; Wen et al., 2025a;b; Fu et al., 2025b; Li et al., 2025a) have been proposed. Beyond binary discrimination, recent efforts (Wen et al., 2025b; Park et al., 2025; Fu et al., 2025b; Li et al., 2025a; Gao et al., 2025) focus on providing artifacts explanations. However, current methods face **two challenges: (1)** Even state-of-the-art (SoTA) multi-modal large language models (MLLMs) struggle to capture human-perceivable artifacts (Wang et al., 2025b). To mitigate this, BusterX++ (Wen et al., 2025b) adopts pure Reinforcement Learning (RL) on large-scale detection dataset, but the resulting model falls into superficial analysis, focusing on coarse features like environment and lighting. **(2)** The task-oriented fine-tuning leads to *mechanical grounded analysis*. Methods like Skyra (Li et al., 2025a) and DeepTraceReward (Fu et al., 2025b) construct grounded Chain-of-Thought (CoT) for Supervised Fine-Tuning (SFT), aiming to inject fine-grained perceptual capacities to base

[†]Project Lead [1]School of Advanced Interdisciplinary Sciences, UCAS [2]MAIS, Institute of Automation, Chinese Academy of Sciences [3]Ant Group [4]Shenzhen Institute of Advanced Technology (SIAT), Chinese Academy of Sciences [5]School of Artificial Intelligence, UCAS. Correspondence to: Jun Wan <jun.wan@ia.ac.cn>.

*Proceedings of the 43rd International Conference on Machine Learning*, Seoul, South Korea. PMLR 306, 2026. Copyright 2026 by the author(s).

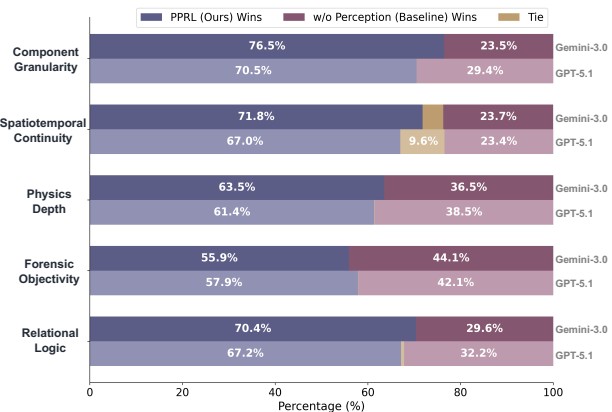

*Figure 2.* **To understand why PPRL enhances detection**, we characterize the model's reasoning behavior across five distinct dimensions, finding that PPRL effectively shapes better reasoning behavior. For instance, model trained with PPRL tends to break down a whole scene into specific objects (i.e., "Component Granularity": 76.5% win rate). Details are provided in Sec. 4.4.

model. However, the resulting model fails to conduct basic fact-based reasoning on AI parodies.

To mitigate these issues, we introduce VIDEOVERITAS, an MLLM-based detector that integrates fine-grained perception and general reasoning. As shown in Figure 1, our approach leverages a two-stage training pipeline. **In the first stage**, rather than performing SFT on large-scale database, we introduce Joint Preference Alignment. Specifically, we construct question-answering (QA) reports to select data with diversified artifacts. To integrate perceptual and reasoning abilities, we incorporate both response-level and video-level alignments, leveraging the base model itself as a strong guidance (e.g., using "anti-label" strategy to probe the intrinsic hallucinations as non-preference). **In the second stage**, we introduce Perception Pretext RL (PPRL). Rather than directly optimizing for AIGC detection task, we adopt (1) general spatiotemporal grounding and (2) self-supervised object counting as perception pretext tasks, which yield promising improvements on detection without requiring label-intensive annotations. As shown in Figure 2, we also empirically demonstrate that PPRL leads to *better reasoning behavior*, which therefore *benefits* detection.

Moreover, videos in existing datasets (e.g., GenVideo (Chen et al., 2024) and GenVidBench (Ni et al., 2025)) primarily come from early models like ModelScope, which suffer from suboptimal temporal consistency. To support more robust evaluation, we present **MintVid**, a light AI-generated video dataset spanning three parts: **(1)** 1.5K highly realistic videos from 6 powerful proprietary models, each including both T2V and TI2V, **(2)** 2K deepfake videos using 3 SoTA public models, and **(3)** a fact-based subset collected from short-video platforms, which could be identified through factual reasoning. Together with available benchmarks, e.g., GenBuster++ (Wen et al., 2025b), the models are evaluated

on a holistic and challenging setting.

Building on MintVid, we reveal that binary detectors like DeMamba (Chen et al., 2024) and RestraV (Internò et al., 2025) struggle to yield satisfactory results. MLLM with cold-start (e.g., Skyra) exhibits notably low recall on fact-based subset, while pure RL (e.g., BusterX++) demonstrates degraded results on those challenging subsets. In contrast, **VIDEOVERITAS** achieves a more balanced performance.

To sum up, our main contributions are:

- We introduce Perception Pretext RL **algorithm**, which leverages simple perception pretext tasks to elevate detection performance. This method can be seamlessly integrated into existing R1-paradigm framework.

- We establish **VIDEOVERITAS**, a **framework** that integrates fine-grained perception and fact-based reasoning for AI-generated video detection, achieving superior results over previous methods across multiple datasets.

- We introduce MintVid, a light yet high-quality **dataset** containing 3K videos from 9 SoTA generators. MintVid facilitates evaluations on three aspects, including general-content, facial and fact-based scenarios.

## 2. Related Work

### 2.1. AI-Generated Video Detection and Datasets

Early studies primarily focused on image-level forgery detection (Ojha et al., 2023; Yan et al., 2024b; Tan et al., 2024; Nguyen et al., 2024; Fu et al., 2025c; Yan et al., 2024a;c; Yang et al., 2025b). With the increasing capabilities of video generation, various efforts are made for video detection (Chen et al., 2024; Bai et al., 2024; Liu et al., 2024b; Ma et al., 2025; Zheng et al., 2025b; Internò et al., 2025; Zhang et al., 2025b; Corvi et al., 2025). For instance, D3 (Zheng et al., 2025b) revealed the discrepancy in the second-order differences of features between real and AI-generated videos, enabling a training-free detection through simple pre-trained encoders. Sharing similar insights, ReStraV (Internò et al., 2025) leverages the statistical discrepancy between pre-trained representations for discrimination. NSG-VD (Zhang et al., 2025b) introduces a novel perspective based on physical conservation principles. Similar to recent "bias-free" approaches (Zhou et al., 2024; Yan et al., 2024b; Kashiani et al., 2025; Chen et al., 2025) in image domain, WaveRep (Corvi et al., 2025) explores unbiased training using video VAE and frequency-level alignment, greatly enhancing detection robustness under H.264 compression. Besides, some methods also target deepfake video detection (Haliassos et al., 2021; Wang et al., 2023; Xu et al., 2023; Han et al., 2025; Kim et al., 2025; Nguyen et al., 2025; Yan et al., 2025), incorporating facial

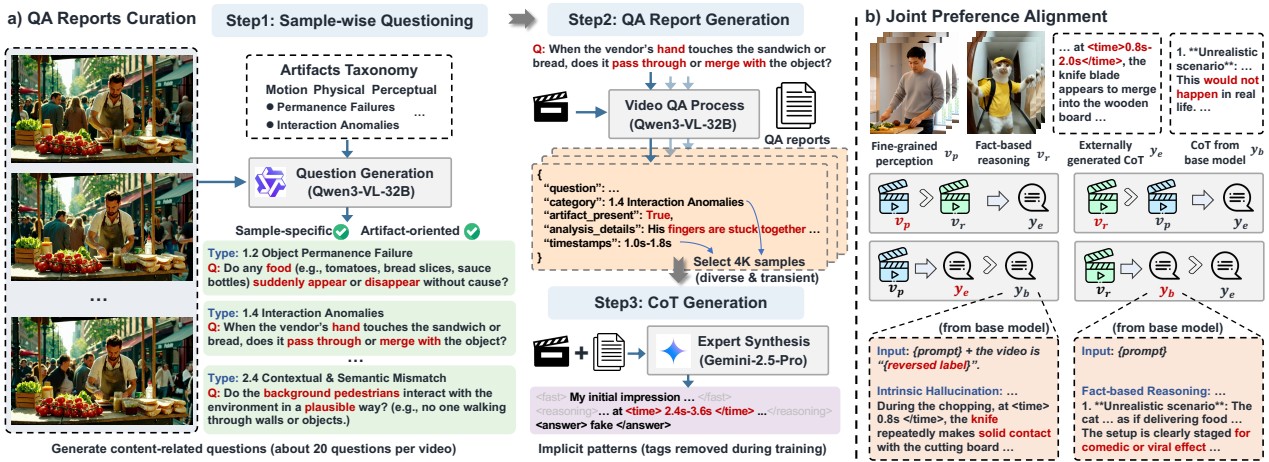

*Figure 3.* **Overview of our Joint Preference Alignment stage. (a)** We generate Question-Answering (QA) Reports to select diverse data and curate high-quality Chain-of-Thought (CoT). It involves generating artifact-oriented questions and creating detailed QA reports. **(b)** Joint DPO constructs preference pairs for both response-level and video-level alignments, leveraging external CoT and the base model's own reasoning to effectively guide the model. The artifacts taxonomy is provided in Appendix B.

priors to achieve robust detection. However, most methods are evaluated on previous datasets, e.g., GenVideo (Chen et al., 2024) and GenVidBench (Ni et al., 2025), where most videos are derived from *outdated* generative models, suffering from limited temporal consistency. Although various benchmarks (Du et al., 2025; Wang et al., 2025a; Huang et al., 2025; Li et al., 2025b) have been proposed recently, they primarily focus on image domain. In this paper, we introduce a video dataset to facilitate more robust evaluation.

## 2.2. Multimodal Large Language Models

**Brief Review of Generic MLLMs.** To encode the temporal information, Qwen2-VL (Wang et al., 2024) employs Multimodal Rotary Position Embedding (M-RoPE). Qwen2.5-VL (Bai et al., 2025b) further incorporates absolute temporal encoding, while Qwen3-VL (Bai et al., 2025a) introduces explicit timestamp tokens, achieving strong performance on generic video benchmarks like VideoMME (Fu et al., 2025a). To adaptively encode long videos, Keye-VL1.5 (Yang et al., 2025a) proposes a slow–fast frames encoding strategy, while VideoL-LaMA3 (Zhang et al., 2025a) introduces a Differential Frame Pruner, greatly improving performance on long video understanding. Although their semantic-level understanding capabilities have greatly advanced, their fine-grained spatiotemporal perception remains limited (Zhao et al., 2025).

**MLLMs for AI-Generated Video Detection.** Leveraging the semantic understanding capabilities of MLLMs, several studies have explored explainable AI-generated video detection (Song et al., 2024; Jiang et al., 2025; Xu et al., 2025; Gao et al., 2025; Fu et al., 2025b; Park et al., 2025; Wen et al., 2025a;b; Li et al., 2025a). DAVID-XR1 (Gao

et al., 2025) and VidGuard-R1 (Park et al., 2025) employ Supervised Fine-Tuning (SFT) using Chain-of-Thought (CoT) generated by Gemini-2.5-Pro and Qwen2.5-VL-72B, respectively. DeeptraceReward (Fu et al., 2025b) introduces "human-perceivable" explainability and involves timestamps and bounding boxes in the reasoning process, while Skyra (Li et al., 2025a) uses 4K manually annotated samples to inject the grounding capacities into base model. In contrast, BusterX++ (Wen et al., 2025b) supposes that low-quality cold starts can affect the generic reasoning ability and therefore adopts pure Reinforcement Learning (RL) on large-scale training data. In this paper, we introduce Perception Pretext RL, which achieves promising gains without the burden of human-annotated AIGC data.

## 3. Method

In this section, we present a detailed description of the two-stage training pipeline and MintVid dataset.

### 3.1. Joint Preference Alignment

**QA reports generation and CoT curation.** As shown in Figure 3, for each video, we first generate various *artifact-oriented* questions that are strongly *related* to its content. Then we use Qwen3-VL-32B (Bai et al., 2025a) to answer each question, forming a comprehensive QA report for each video. Each item contains the type of artifacts and exact timestamps. Such strategy offers two benefits: (1) It enables *independent* and label-agnostic perception of specific artifact, showing fewer hallucinations than directly prompted with Ground-Truth labels. (2) It provides a valuable reference for data sampling. Based on this pipeline, we fil-

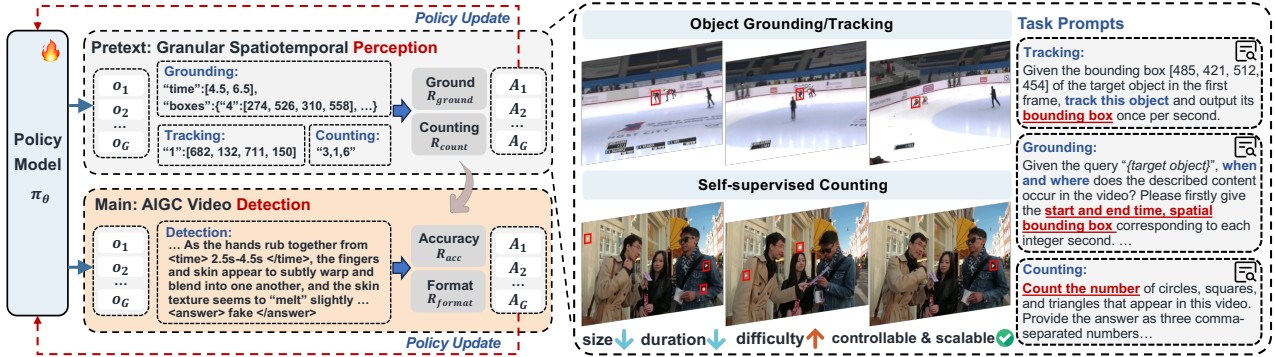

*Figure 4.* **Perception Pretext RL (PPRL).** **Left**: *Perception* is taken as a foundational phase to *detection* . The pretext phase can be implemented with various perception-oriented tasks, e.g., spatiotemporal grounding and object counting. **Right**: Examples of perception pretext tasks. Grounding and tracking data is sampled from OneThinker (Feng et al., 2025). The model is prompted to output exact bounding boxes and timestamps. Self-supervised counting is controllable by adjusting the size and duration of the objects, and the model is required to output the exact quantity of each shape.

ter 4K samples from the database comprising GenBuster-200K (Wen et al., 2025a), RewardData (Fu et al., 2025b), TalkingheadBench (Xiong et al., 2025) and several fact-based videos, where $\boldsymbol{v}_p$ and $\boldsymbol{v}_r$ denote videos that require fine-grained perception and fact-based reasoning, respectively. We utilize Gemini-2.5-Pro to generate CoT based on its own reasoning and QA reports. Following Veritas (Tan et al., 2025), we adopt a structured reasoning design, but remove the explicit "tags" during training, encouraging the model to learn the underlying thinking pattern inherently.

**Response-level alignment.** Suppose previous annotated CoT is denoted as $\boldsymbol{y}_e$, we make full use of the base model itself for more effective alignments: **(1)** We utilize "anti-label" strategy (i.e., insert the reversed label in the prompt) to trigger the base model's *intrinsic* hallucinations, designating these as non-preferred responses. **(2)** For videos requiring factual reasoning, the base model's outputs are directly leveraged as preferred responses. Let $\boldsymbol{y}_b$ denote the CoT generated from base model. For $\boldsymbol{v}_p$ we assume that $\boldsymbol{y}_e$ is preferred over $\boldsymbol{y}_b$ (i.e., $\boldsymbol{y}_e \succ \boldsymbol{y}_b$), whereas for $\boldsymbol{v}_r$, we suppose the opposite preference (i.e., $\boldsymbol{y}_b \succ \boldsymbol{y}_e$). Given the input query $\boldsymbol{x}$, the response-level preference objective is formulated as:

$$u_t(\boldsymbol{x},\boldsymbol{v},\boldsymbol{y}_w,\boldsymbol{y}_l) = \beta\log\frac{\pi_\theta(\boldsymbol{y}_w|\boldsymbol{v},\boldsymbol{x})}{\pi_{\text{ref}}(\boldsymbol{y}_w|\boldsymbol{v},\boldsymbol{x})} - \beta\log\frac{\pi_\theta(\boldsymbol{y}_l|\boldsymbol{v},\boldsymbol{x})}{\pi_{\text{ref}}(\boldsymbol{y}_l|\boldsymbol{v},\boldsymbol{x})},$$
$$\mathcal{L}_{\text{DPO}_t} = -\mathbb{E}_{(\boldsymbol{x},\boldsymbol{v}_p,\boldsymbol{y}_e,\boldsymbol{y}_b)}[\log\sigma(u_t(\boldsymbol{x},\boldsymbol{v}_p,\boldsymbol{y}_e,\boldsymbol{y}_b))] \quad (1)$$
$$-\mathbb{E}_{(\boldsymbol{x},\boldsymbol{v}_r,\boldsymbol{y}_b,\boldsymbol{y}_e)}[\log\sigma(u_t(\boldsymbol{x},\boldsymbol{v}_r,\boldsymbol{y}_b,\boldsymbol{y}_e))],$$

where $\sigma(\cdot)$ denotes sigmoid function.

**Video-level alignments.** To encourage the model to generate preferred outputs based on pure visual information, we involve video-level alignments. Specifically, we encourage the model to output fine-grained analysis $\boldsymbol{y}_e$ based on video $\boldsymbol{v}_p$ (i.e., $\boldsymbol{v}_p \succ \boldsymbol{v}_r$), while outputing factual reasoning $\boldsymbol{y}_b$ when given video $\boldsymbol{v}_r$ (i.e., $\boldsymbol{v}_r \succ \boldsymbol{v}_p$). The video-level

preference objective is computed as:

$$u_v(\boldsymbol{x},\boldsymbol{y},\boldsymbol{v}_w,\boldsymbol{v}_l) = \beta\log\frac{\pi_\theta(\boldsymbol{y}|\boldsymbol{v}_w,\boldsymbol{x})}{\pi_{\text{ref}}(\boldsymbol{y}|\boldsymbol{v}_w,\boldsymbol{x})} - \beta\log\frac{\pi_\theta(\boldsymbol{y}|\boldsymbol{v}_l,\boldsymbol{x})}{\pi_{\text{ref}}(\boldsymbol{y}|\boldsymbol{v}_l,\boldsymbol{x})},$$
$$\mathcal{L}_{\text{DPO}_v} = -\mathbb{E}_{(\boldsymbol{x},\boldsymbol{y}_e,\boldsymbol{v}_p,\boldsymbol{v}_r)}[\log\sigma(u_v(\boldsymbol{x},\boldsymbol{y}_e,\boldsymbol{v}_p,\boldsymbol{v}_r))] \quad (2)$$
$$-\mathbb{E}_{(\boldsymbol{x},\boldsymbol{y}_b,\boldsymbol{v}_r,\boldsymbol{v}_p)}[\log\sigma(u_v(\boldsymbol{x},\boldsymbol{y}_b,\boldsymbol{v}_r,\boldsymbol{v}_p))].$$

The final objective for the Joint DPO is defined as:

$$\mathcal{L}_{\text{J-DPO}} = \mathcal{L}_{\text{DPO}_t} + \mathcal{L}_{\text{DPO}_v}. \quad (3)$$

### 3.2. Perception Pretext RL (PPRL)

As illustrated in Figure 4, rather than directly optimizing for AIGC detection task, we introduce perception pretext tasks. By first encouraging the model to accurately perceive subtle and transient targets, the fine-grained perception capacity is sharpened, benefiting the following detection task. Specifically, we introduce two types of perception tasks: (1) general spatiotemporal grounding and (2) self-supervised object counting.

**General grounding.** This includes spatiotemporal grounding and object tracking. For tracking task, given the bounding box $B_1$ in the first frame, the model is required to predict a sequence of bounding boxes $\{\widehat{B}_i\}_{i=2}^N$ to track the object across the video. Suppose $\{B_i\}_{i=2}^N$ denotes the ground-truth boxes, the reward is measured as the mean Intersection over Union (IoU) over all frames:

$$R_{\text{track}} = \frac{1}{N-1}\sum_{i=2}^N \frac{|\widehat{B}_i \cap B_i|}{|\widehat{B}_i \cup B_i|}. \quad (4)$$

For spatiotemporal grounding, the model is required to predict both the temporal span $\widehat{t}$ of a query and its bounding boxes $\{\widehat{B}_i\}_{i=1}^M$ across the video. Given the ground-truth $t_y$ and $\{B_i\}_{i=1}^M$, the reward is measured according to both the

timestamp and spatial boxes:

$$R_{\text{ground}} = \frac{1}{2} \frac{|\widehat{t} \cap t_y|}{|\widehat{t} \cup t_y|} + \frac{1}{2} \sum_{i=1}^{M} \frac{|\widehat{B}_i \cap B_i|}{|\widehat{B}_i \cup B_i|}. \quad (5)$$

There are abundant samples that are publicly available. In our case, all data are sampled from OneThinker (Feng et al., 2025), with easy samples filtered out, retaining only small and short-duration targets for training.

**Self-supervised object counting.** For a given video, we randomly add a set of objects with randomized attributes (e.g., size, duration and position) to the frames. The model is required to output the counts for each shape category in the video, including circles, squares and triangles. For each shape $s$, the reward $R_{\text{count},s}$ is inversely proportional to the relative prediction error:

$$R_{\text{count},s} = \max(0, 1 - \frac{|\widehat{y}_s - y_s|}{y_s + \epsilon}), \quad (6)$$

where $\widehat{y}_s$ and $y_s$ denotes the predicted and ground-truth counts, respectively. $\epsilon$ is a small constant (e.g., $10^{-6}$) to ensure numerical stability. The final counting reward $R_{\text{count}}$ is averaged from all shapes:

$$R_{\text{count}} = \frac{1}{S} \sum_{s=1}^{S} R_{\text{count},s}. \quad (7)$$

Since the object size and duration are fully controllable, we can generate diverse and challenging training signals at scale. In our case, we randomly select videos from OpenVid-1M (Nan et al., 2024) to construct learning samples. Only with precise spatiotemporal perception can the model correctly count different shapes. This enables efficient scaling without requiring any manual annotations.

**AIGC detection.** For the detection task, we utilize accuracy reward $R_{\text{acc}}$ and format reward $R_{\text{format}}$, where $R_{\text{acc}} = 1$ when the answer is correct and $R_{\text{acc}} = 0$ otherwise. $R_{\text{format}}$ is given when the answer is enclosed in <answer> </answer> tags. The reward for detection task is:

$$R_{\text{detection}} = R_{\text{acc}} + \alpha R_{\text{format}}, \quad (8)$$

where $\alpha = 0.2$ is to control the effect of format reward. Specifically, we use 3K samples for general grounding, 2K samples for self-supervised object counting, and 10K samples (from database in Sec. 3.1) for AIGC detection. The perception task and detection task are performed in a sequential manner. Since our method is suitable for most R1-style algorithms (Liu et al., 2024a), we adopt Group Sequence Policy Optimization (GSPO) (Zheng et al., 2025a) for training. Instead of relying on manually annotated AIGC data or elaborate reward design, we demonstrate that simple RL on perception pretext tasks could achieve promising gains.

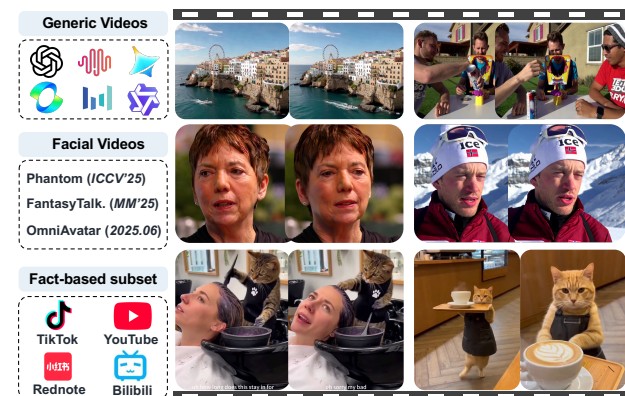

*Figure 5.* **Examples of the MintVid dataset.** MintVid comprises three different parts, which facilitates more complete evaluation.

### 3.3. MintVid Dataset

As shown in Figure 5, MintVid comprises three parts:

**General content videos.** 1.5K videos from 6 powerful proprietary models: Jimeng 3.0 Pro, Seedance 1.0 Pro, Kling 2.5-Turbo, Wan2.5, MiniMax Hailuo 2.3, and Sora2. For each model, we sample 100 meticulous prompts from Video-Verse (Wang et al., 2025d), PhyGenBench (Meng et al., 2024) and VMBench (Ling et al., 2025), and 200 prompts from OpenVid-1M (Nan et al., 2024) captions. Among them, 100 prompts are paired with the corresponding first frame to generate TI2V data. After manual filtering, 1.5K videos are kept. Real videos are sampled from OpenVid-1M.

**Facial videos.** 2K videos from 3 SoTA human-centric video generative models: OmniAvatar (Gan et al., 2025), FantasyTalking (Wang et al., 2025c), and Phantom (Liu et al., 2025), all using their 14B variants. Real videos are sampled from VFHQ (Xie et al., 2022) and HDTF (Zhang et al., 2021). We curate input prompts to yield high-quality videos. Please see Appendix A for more details.

**Fact-based subset.** This part is often overlooked in existing datasets. For videos that violate objective facts, general MLLMs can perform correct reasoning, whereas fine-tuned models may fail due to formulaic analysis. To this end, we collect over 200 videos (including both real and fake) from TikTok, YouTube, and Bilibili, and conduct manual filtering to ensure all videos can be verified using objective facts.

## 4. Experiments

### 4.1. Experimental Setup

**Evaluation Protocols.** In-domain (ID) testing includes test sets of GenBuster-200K (Wen et al., 2025a) and Talking-headBench (Xiong et al., 2025). Out-of-domain (OOD) testing involves existing benchmarks GenBuster++ (Wen et al., 2025b), LOKI (Ye et al., 2024) (using their video set), and

*Table 1.* Performance comparison on video datasets. Accuracy is reported except for D3 (Zheng et al., 2025b), where ∗ means Average Precision (AP) is adopted following the official guideline. The final average performance is calculated by first averaging the results across ID (in-domain), OOD (out-of-domain) and OOD-MintVid, and then taking the average of the three values. The best results are **bolded** and the second best are underlined. More metrics are provided in Appendix D.

| Method | ID | OOD | | | | | | OOD-MintVid | | | | | | | | | | Avg. |
|---|---|---|---|---|---|---|---|---|---|---|---|---|---|---|---|---|---|---|
| | | GenBuster++ | LOKI | Vidu Q1 | Gen-4 | Veo3 | Emu3 | Phantom-14B | OmniAvatar | FantasyTalking | Seedance1.0Pro | Jimeng3.0Pro | Kling2.5-Turbo | Hailuo2.3 | Wan2.5 | Sora2 | Fact | |
| *Small Vision Models* | | | | | | | | | | | | | | | | | | |
| DeMamba | 87.6 | 86.8 | 71.1 | 80.7 | 83.4 | 79.4 | 85.9 | 57.4 | 55.2 | 62.2 | 49.2 | 46.2 | 46.6 | 41.1 | 49.3 | 41.4 | 73.3 | 73.7 |
| D3* (*ICCV'25*) | 49.5 | 46.8 | 39.1 | 55.1 | 89.0 | 66.1 | 90.0 | 54.8 | 36.4 | 44.1 | 61.6 | 48.7 | 51.7 | 78.9 | 50.5 | 54.6 | 32.4 | 55.1 |
| NSG-VD (*NIPS'25*) | 52.4 | 53.3 | 55.1 | 53.4 | 58.2 | 64.3 | 50.7 | 54.3 | 53.8 | 58.5 | 53.8 | 58.0 | 56.0 | 69.4 | 57.0 | 57.8 | 52.8 | 55.1 |
| ReStraV (*NIPS'25*) | 50.7 | 48.9 | 64.8 | 54.8 | 52.8 | 59.8 | 95.2 | 49.6 | 49.3 | 49.9 | 36.9 | 48.4 | 47.2 | 30.2 | 40.9 | 45.4 | 65.0 | 53.2 |
| *Generic MLLMs* | | | | | | | | | | | | | | | | | | |
| Qwen2.5-VL-7B | 54.1 | 59.5 | 49.4 | 58.8 | 53.3 | 60.3 | 99.2 | 50.7 | 50.0 | 49.7 | 63.6 | 49.4 | 51.6 | 53.1 | 55.8 | 50.4 | 87.1 | 57.9 |
| Qwen3-VL-8B | 65.1 | 62.4 | 56.6 | 60.2 | 59.4 | 62.9 | **99.9** | 53.3 | 50.6 | 50.3 | 64.2 | 51.8 | 55.1 | 54.1 | 66.7 | 50.8 | 91.9 | 63.6 |
| InternVL3.5-8B | 51.7 | 59.6 | 44.1 | 56.1 | 53.4 | 56.7 | 99.7 | 51.3 | 50.2 | 49.5 | 62.6 | 49.8 | 49.1 | 50.2 | 53.9 | 54.1 | 91.7 | 56.5 |
| MiMo-VL-7B | 65.8 | 54.3 | 53.6 | 65.5 | 65.8 | 66.0 | 98.8 | 58.4 | 50.1 | 50.6 | 69.7 | 56.7 | 61.7 | 56.1 | 66.0 | 55.4 | 89.9 | 64.8 |
| Keye-VL-1.5-8B | 52.8 | 62.4 | 60.4 | 58.3 | 56.0 | 62.4 | 94.7 | 51.3 | 51.1 | 53.3 | 59.7 | 50.5 | 51.0 | 52.8 | 52.9 | 51.8 | 80.4 | 58.0 |
| GLM-4.5V | 61.7 | 59.4 | 49.4 | 56.5 | 56.6 | 62.4 | 97.1 | 57.9 | 52.2 | 50.5 | 63.2 | 48.9 | 49.2 | 49.8 | 58.3 | 47.9 | 88.2 | 60.6 |
| Qwen3-VL-235B-A22B | 66.7 | 64.6 | 57.9 | 57.4 | 65.7 | 63.4 | 98.5 | 54.1 | 50.9 | 51.2 | 65.8 | 49.1 | 32.3 | 49.8 | 63.5 | 50.5 | 93.6 | 63.6 |
| Gemini-2.5-Pro | 80.5 | 71.8 | 70.2 | 70.8 | 76.0 | 70.4 | 93.3 | 57.8 | 55.8 | 55.6 | 69.4 | 60.2 | 65.6 | 62.7 | 65.7 | 60.9 | 88.3 | 73.4 |
| Gemini-3-Pro-Preview | 79.4 | 82.5 | 74.9 | 83.1 | 76.3 | 85.7 | 89.3 | 65.5 | **65.3** | 61.5 | 81.4 | 72.1 | 73.1 | 74.3 | 73.1 | 63.5 | 91.4 | 77.8 |
| *MLLM-based Video Forgery Detectors* | | | | | | | | | | | | | | | | | | |
| BusterX++ | 77.1 | 79.6 | 70.9 | 53.0 | 89.9 | 76.2 | 98.9 | 62.3 | 48.6 | 47.6 | 71.8 | 71.7 | 82.3 | 70.4 | **87.3** | 58.7 | 88.8 | 74.7 |
| Skyra-RL | 52.1 | 55.7 | 37.7 | 52.9 | 59.1 | 55.9 | 51.5 | 50.1 | 49.9 | 49.9 | 63.4 | 51.7 | 53.5 | 51.7 | 59.6 | 50.6 | 51.9 | 52.5 |
| **VIDEOVERITAS (ours)** | **93.1** | **91.4** | **78.1** | **93.6** | **96.7** | **94.5** | 99.4 | **79.0** | 56.2 | **84.5** | **86.6** | **80.8** | **86.0** | **85.8** | 86.3 | **67.6** | **96.1** | **88.8** |

high-quality subsets (e.g., Gen-4) from VBench-2 (Zheng et al., 2025c) and MVAD (Hu et al., 2025). MintVid is adopted as a challenging OOD testing. 4 SoTA binary detectors are trained on our dataset using their original preprocessing. Both public and proprietary MLLMs are evaluated using CoT prompting. We also evaluate 2 recent MLLM-based video forgery detectors, including BusterX++ (Wen et al., 2025b) and Skyra-RL (Li et al., 2025a).

**Implementation Details.** We implement the VIDEOVERITAS with Qwen3-VL-8B (Bai et al., 2025a). For Joint DPO, we train the model for 3 epochs using LoRA (Hu et al., 2022) (rank=64, $\alpha$=128), with a learning rate of $5 \times 10^{-5}$ and batch size of 64. For PPRL, the model is trained for 2 epochs with the same LoRA setting, and the learning rate is set to $1 \times 10^{-5}$ with a batch size of 32. The temperature is set to 1.0. Each epoch contains both perception and detection tasks. The videos are sampled at 3 FPS for up to 5 seconds, keeping their native resolutions.

### 4.2. Main Results

**Comparisons to binary detectors.** As shown in Table 1, recent statistical-based methods (Zheng et al., 2025b; Internò et al., 2025) are effective on certain subsets (e.g., 90.0% and 95.2% on Emu3), but their performance collapses to near-chance levels on most contemporary benchmarks, and even fails on in-domain subsets. This suggests that statistical discrepancies may not be a robust criterion for more realistic videos. DeMamba (Chen et al., 2024) performs well on previous datasets like GenBuster++, but suffers a sharp

performance drop on the more challenging MintVid dataset. In contrast, VIDEOVERITAS exhibits certain advantages, achieving 15.1% averaged gains over the previous best.

**Comparisons to SoTA MLLMs.** As shown in Table 1, although generic MLLMs demonstrate strong performance on the fact-based subset, their performance on other datasets remains limited, exhibiting low recall (presented in Table 10). Notably, Gemini-3.0-Pro-Preview outperforms most open-source models and exhibits a *clear lead* on MintVid. Its average performance even surpasses finetuned models like DeMamba and BusterX++, indicating that *a substantial gap still exists* between public and proprietary models. Moreover, our approach achieves an average improvement of 25.2% over base model, and it also outperforms those MLLMs with similar parameter scale, demonstrating the effectiveness of our training strategy.

**Comparisons to MLLM-based detectors.** As shown in Table 1, BusterX++ achieves 74.7% average accuracy, but shows suboptimal performance on MintVid, especially the facial subsets. Skyra-RL proves largely ineffective across most datasets. We suppose this may stem from the strict preprocessing applied to its training and testing data (e.g., 256p resizing), which could bias the model toward exploiting minor inconsistencies. In contrast, VIDEOVERITAS outperforms them on (1) existing datasets like GenBuster++ (+11.8%) and LOKI (+7.2%), (2) perception-heavy datasets like Jimeng3.0-Pro (+9.1%), and (3) fact-dependent subset, achieving a more balanced and robust detection capability.

*Table 2.* Ablations on the type of perception pretext tasks. 5K perception data are taken as default setting.

| Pretext Tasks | | | ID | | OOD | | MintVid | |
|---|---|---|---|---|---|---|---|---|
| SSL | G-G | A-G | Acc | F1 | Acc | F1 | Acc | F1 |
| ✗ | ✗ | ✗ | 92.9 | 93.4 | 91.2 | 92.1 | 77.8 | 73.3 |
| ✓ | ✗ | ✗ | 91.4 | 92.1 | 91.3 | 92.3 | 80.4 | **78.0** |
| ✗ | ✓ | ✗ | 92.3 | 92.8 | 91.1 | 92.2 | 79.8 | 76.7 |
| ✗ | ✗ | ✓ | 92.7 | 93.2 | 91.2 | 92.4 | 79.5 | 76.9 |
| ✓ | ✓ | ✗ | 93.1 | 93.5 | **92.3** | **93.4** | **80.9** | 77.3 |
| ✓ | ✗ | ✓ | 92.9 | 93.3 | 92.0 | 92.9 | 79.8 | 77.0 |
| ✗ | ✓ | ✓ | **93.5** | **93.9** | 91.3 | 92.2 | 79.4 | 76.0 |
| ✓ | ✓ | ✓ | 92.7 | 93.2 | 91.3 | 92.2 | 78.6 | 75.0 |

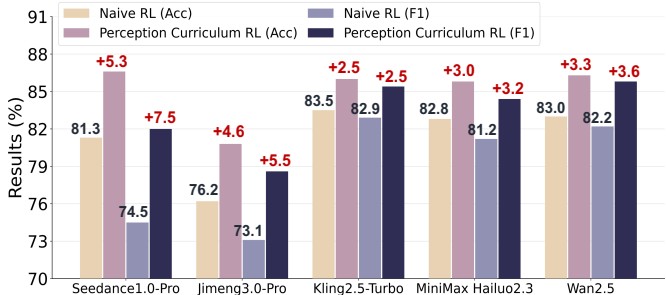

*Figure 6.* Improvements of the proposed PPRL. "SSL+G-G" is adopted in our method.

*Table 3.* Ablations on the training stages. † means using format reward (Li et al., 2025a) to incentivize grounded reasoning.

| Method | ID | OOD | MintVid | |
|---|---|---|---|---|
| | | | Others | Fact |
| Base | 65.1 | 66.9 | 55.2 | 91.9 |
| + SFT | **91.9** | 82.5 | 68.3 | 90.7 |
| + J-DPO | 91.4 | **83.0** | 69.1 | **94.6** |
| − DPO$_v$ | 91.3 | 82.7 | **69.1** | 92.9 |
| − DPO$_t$ | 89.2 | 80.9 | 66.3 | 88.3 |
| + Pure RL | 94.2 | 88.3 | 69.1 | **96.1** |
| + Pure RL† | **94.5** | 90.2 | 70.9 | 87.5 |
| + J-DPO + RL | 93.1 | **92.3** | 79.2 | **96.1** |

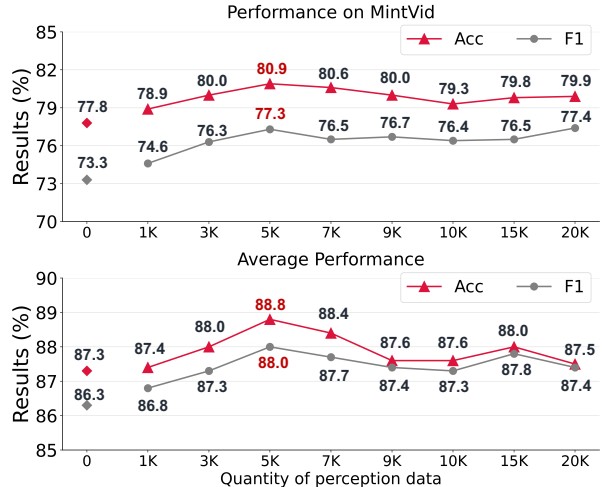

*Figure 7.* Ablations on the quantity of perception data. Accuracy and F1 are reported on MintVid (upper) and Average (lower). Quantity of "0" denotes the baseline.

## 4.3. Ablation Studies

**Ablations on the training stages.** As shown in Table 3, SFT and Joint DPO (J-DPO) is trained on 4K samples. "J-DPO+RL" is further trained on 10K AIGC data and 5K perception data. Pure RL is trained on 20K samples, which ensures similar data scale to investigate the effect of cold-start. Specifically, we draw the following observations: **(1)** Supervised Fine-Tuning (SFT) achieves great improvements (e.g., +13.1% on MintVid-Others) but degrades the performance on Fact subset. In contrast, J-DPO effectively *integrates* the overall capacities, boosting accuracy on Fact subset to 94.6%. **(2)** Video-level alignment (DPO$_v$) is beneficial to synergizing the fact-based reasoning ability, while response-level alignment (DPO$_t$) is critical to overall performance. **(3)** The performance of pure RL on challenging subsets remains limited (e.g., 69.1% vs. 79.2% compared to ours), indicating the necessity of cold-start.

**Ablations on PPRL: the type of perception tasks.** Generally, the perception tasks can be categorized into three types with increasing annotation cost: **(1) Self-Supervised Learning (SSL)**, where we use object counting as a zero-cost pretext for foundational perception. **(2) Generic Grounding (G-G)**, where abundant public data can be utilized. **(3) Artifact Grounding (A-G)**, which is the most specialized and costly task. As shown in Table 2, each perception task independently improves the performance. Combining SSL and G-G yields further enhancements, achieving substantial gains on MintVid dataset as shown in Figure 6, suggesting that the *perceptual acuity gained from these tasks* provides

a better foundation for detection. Interestingly, further combining A-G degrades the performance, which we suppose is due to the quality of A-G annotations. For A-G, we use data sampled from Molmo2 (Clark et al., 2026), which is coarse-grained (e.g., the entire person is masked out even only the arm deforms), making them less compatible with the fine-grained signals from SSL and G-G. We suppose the results might be different if *higher-quality artifact grounding annotations* are available. Overall, without any domain-specific annotations, using SSL and G-G as perceptual pretext tasks yields promising improvements. Detailed results can be found in Appendix Table 11.

**Ablations on PPRL: the quantity of perception data.** As shown in Figure 7, even introducing a small amount of perception data yields a measurable performance gain (i.e., +1.1% Acc and 1.3% F1 on MintVid with 1K perception data). As the amount increases, performance improves steadily. Excessive perception data does not further enhance detection accuracy, but it remains consistently better than the baseline. A scale around 5K is efficient and effective to deliver the best performance improvement.

*Table 4.* Difficulty of the SSL task. (·,·) denotes the range of values ("*the number of pixels per side*" for "Size" and "*seconds*" for "Duration"). "Baseline" is the model trained exclusively on AIGC detection task.

| Difficulty | Size | Duration | ID | | OOD | | MintVid | |
|---|---|---|---|---|---|---|---|---|
| | | | Acc | F1 | Acc | F1 | Acc | F1 |
| Baseline | - | - | 92.9 | 93.4 | 91.2 | 92.1 | 77.8 | 73.3 |
| Easy | (120, 240) | (3s, 5s) | 92.5 | 92.9 | 91.3 | 92.2 | 79.5 | 76.3 |
| Medium | (40, 120) | (2s, 4s) | 92.9 | 93.1 | **92.5** | 93.3 | 79.6 | 76.0 |
| Hard | (20, 40) | (1s, 3s) | **93.1** | **93.5** | 92.3 | **93.4** | **80.9** | **77.3** |
| Super-hard | (15, 20) | (0.2s, 1s) | 92.7 | 93.2 | 91.3 | 92.4 | 80.8 | 77.1 |

*Table 5.* Attempts of batch-level perception integration. ‡ means removing the token normalization term when computing loss.

| Method | ID | | OOD | | MintVid | |
|---|---|---|---|---|---|---|
| | Acc | F1 | Acc | F1 | Acc | F1 |
| Baseline | 92.9 | 93.4 | 91.2 | 92.1 | 77.8 | 73.3 |
| Batch-level | 84.6 | 86.4 | 88.4 | 88.8 | 76.8 | 74.1 |
| Batch-level‡ | 89.7 | 90.7 | 91.4 | 92.1 | 79.2 | 76.4 |
| Phase-level | **93.1** | **93.5** | **92.3** | **93.4** | **80.9** | **77.3** |
| Δ Batch-level‡ | +3.4 | +2.8 | +0.9 | +1.3 | +1.7 | +0.9 |

*Figure 8.* **Comparisons between VIDEOVERITAS and existing MLLMs.** A small car on the road distorts and disappears.

## 4.4. Further Analyses

**Difficulty of perception task.** As mentioned in Sec. 3.2, the difficulty of SSL task is controllable. We investigate different settings by adjusting the size and duration of the target objects, where smaller size and shorter exposure times make the task more challenging. As shown in Table 4, increasing the difficulty is beneficial to the detection performance, with the "Hard" setting achieving the best results. Actually, the "Hard" setting is already *non-trivial* for humans to perceive, yet the model can still be trained to count them accurately. This suggests that the perceptual capacity of *machines* may surpass that of *humans*, but remains underexploited.

**Why could perception pretext task benefit detection.** We provide more details about Figure 2. To facilitate *objective* evaluation, we (1) define five different dimensions and (2) perform pairwise evaluation rather than absolute scoring, where we randomly select 2K samples (1.5K fake and 0.5K real) that both the baseline and PPRL trained model yield correct answers. Gemini-3.0-Pro-Preview and GPT-5.1 are adopted for judgment and the results can be "win", "lose" or "tie". The results show that PPRL trained model tends to

analyze *fine-grained entities*, pay more attention to the *object relation* and performs better at *tracking changes*, which consequently benefits the detection. Evaluation prompts and the definition of the dimensions are listed in Appendix C.

**Batch-level integration vs. Phase-level learning.** Besides phase-level learning, we also explored batch integration, where perception and detection task are paired within each mini-batch and trained jointly. As shown in Table 5, it is non-trivial to directly perform batch integration. The outputs of detection task are significantly longer than perception counterparts, which limits effective gradients for detection under standard GSPO setting. We modified the loss computation to balance gradient allocation, which results in certain gains over baseline. However, phase-level learning remains more effective and is also simpler to apply in practice.

**Case studies.** As compared in Figure 8, the base model fails to capture fine-grained visual details. BusterX++ delivers correct answer, but the reasoning is superficial, focusing on elements like the overall context. Skyra-RL conducts template-like analysis and exhibits suboptimal perception on the high-quality video. In contrast, **VIDEOVERITAS** cor-

rectly identifies the distortion of the tiny car, which demonstrates superior fine-grained perception capacities.

## 5. Conclusion

In this paper, we introduced **VIDEOVERITAS**, a framework that integrates fine-grained perception and fact-based reasoning for AI-generated video detection. We introduce PPRL, which improves perceptual capacities by incorporating simple perception pretext tasks. We also introduce MintVid, a challenging dataset that contains three evaluation aspects. Extensive experiments demonstrate the effectiveness of **VIDEOVERITAS**, highlighting the value of learning foundational perceptual skills for complex detection tasks.

## Impact Statement

This paper presents work whose goal is to advance the field of Machine Learning. There are many potential societal consequences of our work, none which we feel must be specifically highlighted here.

## Acknowledgments

This work was supported in part by the New Generation Artificial Intelligence-National Science and Technology Major Project (No. 2025ZD0123501), the Beijing Natural Science Foundation JQ23016, the Chinese National Natural Science Foundation Projects 62476273 and 62406320, the Science and Technology Development Fund of Macau Project 0044/2024/AGJ, and 0084/2024/RIB2, and Ant Group Research Fund.

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

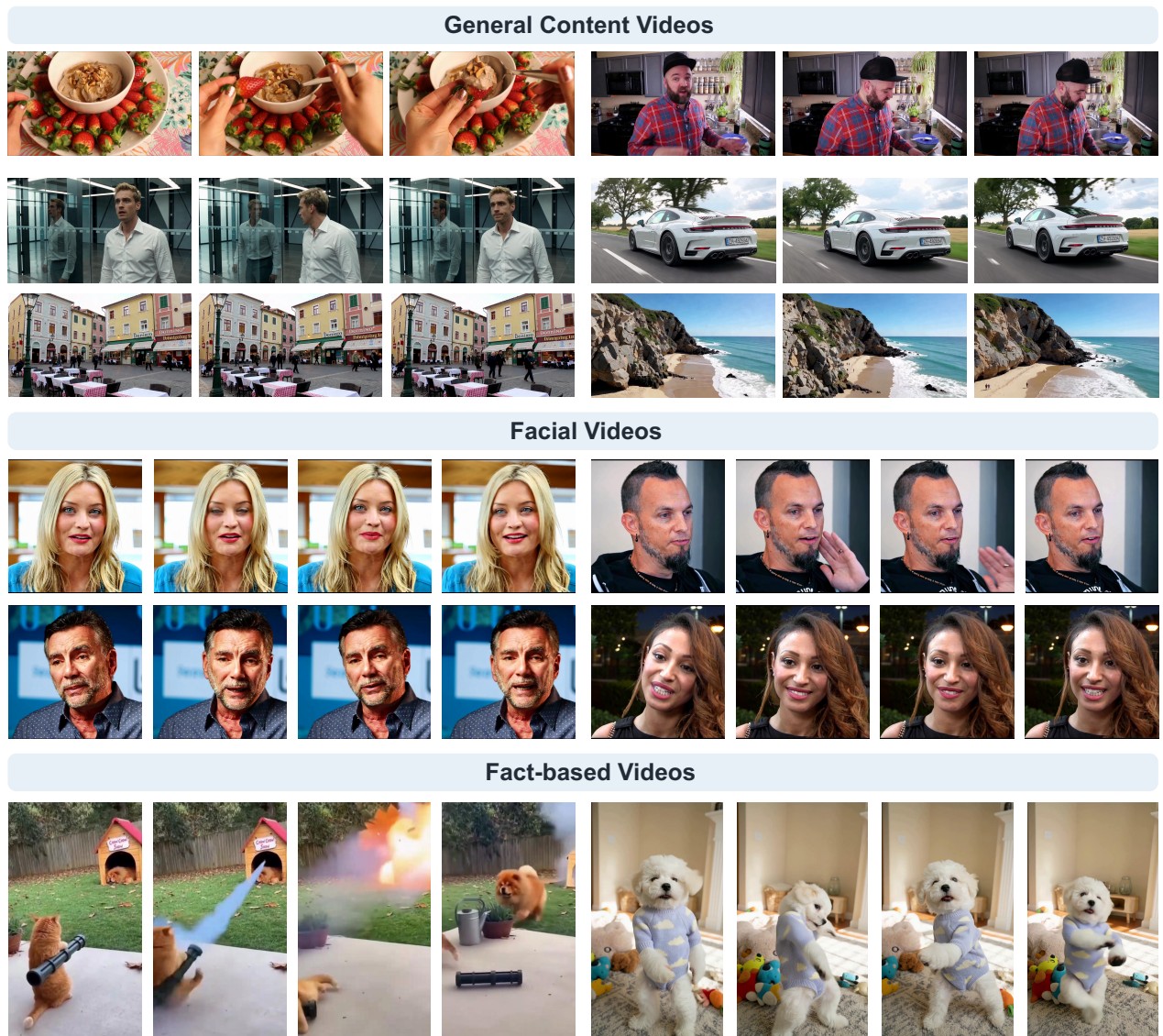

*Figure 9.* Visual examples of MintVid dataset.

## A. More Details of MintVid Dataset

**Generation pipeline of facial videos.** To ensure the quality of the generated videos, we carefully curate the input prompts and conditional images. For the conditional images, we manually select frames from VFHQ (Xie et al., 2022) and HDTF (Zhang et al., 2021), ensuring the clearity of selected frames. For input prompts, we follow the commonly adopted structure, i.e., "**detailed caption of the first frame + action control + background description**". Specifically, we use Qwen2.5-VL-32B (Bai et al., 2025b) to generate detailed caption for the input image. Then we curate about 20 action prompts, including control over facial expressions and gestures, e.g., "slowly turns head from left to right, then briefly dips chin downwards before returning to a neutral position". We further curate about 20 background descriptions, covering different camera controls and environment settings, e.g., "The camera is fixed, creating a stable frame that emphasizes the subject's presence. The lighting is soft and the background is tastefully blurred.". Finally, we utilize Qwen2.5-VL-32B (Bai et al., 2025b) to check the plausibility of the prompts, e.g., whether the subject conflicts with the person's gender in the image and whether background control contradicts the image content. An example of input prompts is provided in Figure 10.

**Visual examples of MintVid dataset.** The key attribute of MintVid dataset is that it considers three distinct aspects, aiming to facilitate more complete evaluation. As mentioned in Sec. 3.3, general content videos are generated using 6 proprietary models. Facial videos are generated using 3 specialized public models. Fact-based videos are collected from

---

**An example of input prompt for facial video generation**

A man is shown in a professional setting, likely within a governmental or official building. He is wearing a dark suit, a striped dress shirt, and a patterned tie. His hair is light-colored and neatly combed, and he is wearing rectangular-framed eyeglasses. The background features ornate columns and an American flag, indicating a significant or ceremonial context. He appears to be speaking or giving a statement, though his expression is calm and focused. He brings his left hand to his forehead as if in thought. The lighting mimics the soft, warm glow of the golden hour, casting a flattering light on the man. The ornate columns and American flag remain distinct in the background.

---

*Figure 10.* An example of input prompt for facial video generation.

online platforms and manually filtered. We provide more visual examples in Figure 9.

**Summary of training data.** To avoid ambiguity caused by rounded data counts, we summarize the training data used in different stages in Table 6. The J-DPO stage contains 4,219 samples, while the RL stage contains 10,789 AIGC samples and 5,000 perception samples, resulting in 20,008 training samples in total. The Pure RL baseline follows a comparable 20K training scale.

*Table 6.* Summary of training data used in different stages.

| Stage | Data Type | Source | Samples |
|---|---|---|---|
| J-DPO | AIGC detection videos | GenBuster-200K, RewardData, TalkingheadBench, OpenVid, and fact-based videos from YouTube | 4,219 |
| RL (perception stage) | Perception pretext data | Grounding data from OneThinker and counting data constructed from OpenVid | 5,000 |
| RL (detection stage) | AIGC detection videos | Same sources as J-DPO | 10,789 |
| **Total** | - | - | **20,008** |
| Testing Data | - | GenBuster++, LOKI, VBench-2, MVAD, MintVid | - |

## B. Artifacts Taxonomy

To enable more fine-grained analyses, we summarize a taxonomy of common artifacts observed in AI-generated videos, covering 3 main perspectives and 11 detailed aspects.

**Motion-level Anomalies:**

- **Unnatural Kinematics and Trajectories:** The movement of an object or being defies physical dynamics or biomechanics in its speed, acceleration, or path.

- **Object Permanence Failure:** An object's existence or individuality is violated, causing it to suddenly appear from nowhere, abruptly vanish without reason, illicitly merge with other distinct entities (e.g., two people blending into a single mass), or spontaneously split into multiple copies.

- **Structural Integrity Failure:** A single object, while maintaining its presence, fails to preserve its own physical form, causing its structure or identifying details to illogically warp, stretch, tear, or distort during motion. The object behaves as if made of the wrong material (e.g., a rigid pole bending like rubber).

- **Interaction Anomalies:** Objects violate their physical properties upon contact, or the scene's spatial coherence breaks down during camera movement.

- **Biological Motion Irregularity:** The natural, subtle dynamic features of living beings, such as blinking, breathing, or micro-expressions, are absent, stiff, or unnaturally timed.

**Physical-level Anomalies:**

- **Inconsistent Lighting and Optics:** Shadows, highlights, or reflections behave inconsistently with the scene's established light sources, object geometry, or movement.

- **Causality and Property Violation:** The sequence of events defies cause-and-effect logic, or an object's behavior contradicts its inherent physical properties.

- **Flawed Material Simulation:** The dynamic simulation of complex, non-rigid materials like cloth, smoke, fire, or water appears unrealistic and lacks natural behavior.

- **Contextual and Semantic Mismatch:** The combination of scenes, objects, and actions is logically or commonsensically contradictory, even if individual elements are rendered realistically.

**Perceptual-level Anomalies:**

- **Texture and Surface Instability:** An object's surface texture exhibits a high-frequency flicker, "crawling" pattern, or shimmer that is disconnected from its motion.

- **Definition and Clarity Fluctuation:** The sharpness and detail level of a region abruptly and illogically degrade or change without a corresponding cause.

## C. Extended Details of Methodology: Prompts and Implementation Details

**Details of reasoning behavior analysis (Figure 2).** We adopt 5 objective dimensions to investigate the model's reasoning behavior:

- **Component Granularity:** This dimension evaluates whether the assistant describes the scene as a whole or deconstructs it into specific objects and their fine-grained components (e.g., specific limbs, lens edges, or individual textures).

- **Spatiotemporal Continuity:** This dimension evaluates the assistant's ability to anchor its observations to precise temporal markers (timestamps) and spatial locations. It looks for how well the analysis "tracks" changes over a specific timeline.

- **Physics Depth:** This dimension evaluates the extent to which the assistant uses principles of physics (optics, mechanics, biology) to explain anomalies, rather than simply stating that something "looks wrong".

- **Forensic Objectivity:** This dimension evaluates the shift from subjective, impression-based language (e.g., "uncanny", "beautiful") to objective, evidence-based descriptions (e.g., "static texture overlay", "non-uniform deformation").

- **Relational Logic:** This dimension assesses the analysis of how different elements in the report interact with one another (e.g., the relationship between a moving object and its shadow, or the reaction of a surface to a force).

The evaluated samples are correctly answered by both models, which avoids the answer bias. For each dimension, we compare the outputs of baseline (trained without perception data) and our model (trained with PPRL). To mitigate potential bias, we utilize two powerful model, i.e., Gemini-3.0-Pro-Preview and GPT-5.1 to conduct pairwise comparison. The input prompts are presented in Figure 19. Specifically, the judging models are required to provide a brief analysis and judgment: model A is better, model B is better or the two models are tied. Then, we adopt the win rate as a straightforward and unbiased evaluation of the performance. An example of the judgment is shown below.

**Prompts of VIDEOVERITAS.** Our framework involves perception tasks and detection task, and we do not require careful curation of input prompts. For detection, the system prompt and input prompt are presented in Figure 12 and Figure 13. For perception, the system prompt is shared as presented in Figure 14. The input instructions are different among different sub-tasks, as shown in Figure 15, Figure 16, Figure 17 and Figure 18.

**More implementation details.** Our model is trained with 8 PPUE GPUs based on ms-swift (Zhao et al., 2024). The theoretical peak computational capacity (TFLOPS) of one PPUE GPU is roughly half of an NVIDIA A100 GPU, and each PPUE GPU has 96GB VRAM. Suppose the training dataset of RL stage is denoted as $\mathcal{D}$, the objective function of GSPO is formulated as:

$$\mathcal{L}_{\text{GSPO}} = -\mathbb{E}_{(\boldsymbol{v},\boldsymbol{y})\sim\mathcal{D},\{\boldsymbol{o}_i\}_{i=1}^G \sim \pi_{\theta_{\text{old}}}(\cdot|\boldsymbol{v})} \frac{1}{G}\sum_{i=1}^{G}\frac{1}{|\boldsymbol{o}_i|}\sum_{t=1}^{|\boldsymbol{o}_i|}\left[\min\left(r_i(\theta)A_{i,t}, \text{clip}(r_i(\theta), 1-\epsilon_1, 1+\epsilon_2)A_{i,t}\right)\right], \quad (9)$$

---

**Judgment Output of "Component Granularity"**

"analysis": "Assistant A demonstrates superior granular analysis by deconstructing the scene into specific components with detailed observations. **It isolates and analyzes specific elements** such as **"small bubbles"** (noting their spherical shape and static nature), distinct light reflections (described as solid, glowing shapes), and the specific "sun" object (a glaring red dot) versus the color of the light on the water. **In contrast, Assistant B focuses on broader features** like the general "oil slick", "edges", and "overall scene", lacking the fine-grained decomposition found in A.",
"judgment": "[[A]]"

---

*Figure 11.* Example of jugement output to "Component Granularity". The judging model is Gemini-3.0-Pro-Preview.

where importance ratio $r_i$ is defined based on sequence likelihood:

$$r_i(\theta) = \left( \frac{\pi_\theta(\boldsymbol{o}_i|\boldsymbol{v})}{\pi_{\theta_{\text{old}}}(\boldsymbol{o}_i|\boldsymbol{v})} \right)^{\frac{1}{|\boldsymbol{o}_i|}},\tag{10}$$

and the advantage is calculated based on group estimation:

$$A_{i,t} = \frac{R_i - \text{mean}(\{R_1, \ldots, R_G\})}{\text{std}(\{R_1, \ldots, R_G\})}.\tag{11}$$

Specifically, $G$ is set to $4$ in our method. $\epsilon_1$ and $\epsilon_2$ are set to $3 \times 10^{-4}$ and $4 \times 10^{-4}$, respectively. In the ablations of Table 5, we found that it is non-trivial to perform batch-level perception integration with the above formulation, since the objective is normalized by the output length in Eq. 9. To balance the gradient between different tasks, we remove the normalization term $\frac{1}{|\boldsymbol{o}_i|}$ in Eq. 9. This greatly elevates the performance, which is superior than the baseline, demonstrating that incorporating perception learning is beneficial to our detection task.

## D. More Experimental Results

**More metrics.** As presented in Table 10, we further report recall and F1 (with "fake" taken as positive category). Specifically, NSG-VD exhibits high recall on several subsets, while the overall accuracy is near-chance level. RestraV suffers from low recall on most datasets. Zero-shot MLLMs tend to predict videos as real, exhibiting low recall on most datasets. However, Gemini-2.5-Pro and Gemini-3-Pro-Preview show *clear and decisive leads* in recall, indicating that their perceptual capabilities remain significantly stronger than those of open-source models. Notably, most models achieve near-perfect results on Emu3, which is collected from MVAD (Hu et al., 2025). The reason may come from two aspects: (1) most videos in Emu3 are in an anime style, which are generally easy for MLLMs to judge, and (2) the distribution produced by autoregressive generators like Emu3 may be quite different. BusterX++ performs well on most datasets, but exhibits obvious shortcomings on datasets like OmniAvatar and FantasyTalking. Skyra-RL fails on most datasets, achieving only 6.9% averaged recall, which as we mentioned in the main text, may be due to its controlled training procedure. Overall, VIDEOVERITAS yields more balanced performance, achieving superior recall and F1 score over previous methods.

**More comparisons.** We provide more comparisons of reasoning outputs in Figure 21, Figure 22, Figure 23, Figure 24 and Figure 25. Although Skyra-RL can conduct grounded analysis by examining specific regions, their analyses are mechanical and imprecise, which leads to failures on those fact-based videos, e.g., AI parodies in Figure 25. BusterX++ gets correct answers on most samples, but the *reasoning content is superficial*, mainly focus on the macro-level semantic concepts like clothing and lighting. In contrast, VIDEOVERITAS demonstrates superior fine-grained perceptual capacity and fact-based reasoning ability. For instance, in Figure 21 upper, previous methods all fail to perceive the distortion of the person's feet when entering the boat, while VIDEOVERITAS precisely captures this artifact. In Figure 21 lower, VIDEOVERITAS correctly captures the object permanence failure, where a bottle emerges suddenly and the materials in the bottle have unexpectedly changed.

**Perception capacity of strong MLLMs.** A common concern is that those proprietary MLLMs had been trained on extensive perception data, why could thousands of samples in our PPRL bring improvements? The key observation is that these powerful models still fall short on highly difficult samples. As shown in Table 7, we present the performance on our

sampled tracking and counting tasks. Gemini-3.0-Pro-Preview struggles under challenging settings such as Counting-Hard. Meanwhile, open-source MLLMs still lag behind proprietary models on perception-heavy tasks. These results suggest that large-scale generic pretraining remains insufficient for granular spatiotemporal perception, while PPRL provides targeted signals for hard, localized, and temporally precise observations.

**Reasoning-quality evaluation.** As shown in Table 8, we conduct both MLLM-as-a-Judge and human judgment to evaluate the quality of model explanations. For MLLM-as-a-Judge, we randomly sample 1,000 videos and their reasoning outputs (800 fake and 200 real), and use Gemini-3.0-Pro-Preview for pairwise comparison. For human judgment, we randomly sample 200 videos (150 fake and 50 real), forming 600 pairwise comparisons among three models. Annotators are asked to compare explanations from three perspectives: perception accuracy, forensic objectivity, and logical coherence. Following prior practice, we compute the final ranking with Elo rating. The results show that VIDEOVERITAS achieves the highest Elo score under both automatic and human protocols, which is consistent with the qualitative comparisons. The human-judgment platform screenshot is provided in Figure 20.

**Performance of image-level detection.** Although VIDEOVERITAS is trained exclusively on video data, its core capacities, i.e., fine-grained perception and fact-based reasoning, are also useful for image-level detection. As shown in Table 9, VIDEOVERITAS achieves competitive results on both GenBuster++ (Wen et al., 2025b) and Chameleon (Yan et al., 2024a) benchmarks, indicating that the learned perceptual and reasoning skills can transfer beyond the video setting.

**Brief summary of explored attempts.** We briefly summarize several unsuccessful attempts. (1) For batch-level integration, we explored conditional reward designs for tight coupling, but it did not achieve improvements comparable to phase-level learning. (2) We tried more forms of SSL like temporal anomaly grounding (e.g., generated by shuffling local clip), but they were less effective than spatiotemporal tasks like object counting. (3) We also tried more detailed reward designs for detection task, e.g., explicitly encouraging grounding formats like `<time> </time>`, which were found to be vulnerable to reward hacking. However, we suppose the results drawn from such heuristic designs may *differ* when high-quality annotations are available.

*Table 7.* Perception and detection performance of strong MLLMs. Counting, Tracking, and MintVid accuracy are reported. "M" stands for Medium, and "H" represents Hard.

| Model | Count-M | Count-H | Tracking | MintVid |
|---|---|---|---|---|
| Gemini-3.0-Pro-Preview | 79.4 | 50.8 | 42.4 | 72.1 |
| Qwen3-VL-235B-A22B | 79.5 | 25.5 | 36.5 | 56.1 |
| GLM-4.5V | 75.1 | 34.7 | 39.8 | 56.6 |
| VIDEOVERITAS (8B) | **93.5** | **72.9** | **48.3** | **80.9** |

*Table 8.* Elo scores for reasoning-quality evaluation. Higher scores indicate better explanation quality. Gemini-3.0-Pro-Preview is adopted for MLLM-as-a-Judge.

| Method | MLLM-as-a-Judge | Human-Judge |
|---|---|---|
| Skyra | 784.30 | 724.50 |
| BusterX++ | 963.29 | 1089.21 |
| VIDEOVERITAS | **1252.41** | **1186.30** |

*Table 9.* Transferability to image-level detection. The reported metrics are "Acc/Recall/F1".

| Method | Training | GenBuster++ | Chameleon |
|---|---|---|---|
| Qwen3-VL-8B | – | 62.1/28.9/43.4 | 72.0/57.2/63.7 |
| FakeVLM | Image | 55.5/**86.5**/66.0 | 62.9/**99.7**/69.7 |
| BusterX++ | Image+Video | 78.1/75.7/77.6 | 76.1/80.5/74.3 |
| VIDEOVERITAS | Video | **86.4**/76.2/**84.9** | **79.8**/88.9/**79.1** |

# E. Limitation and Future Work

One limitation of our work is that the perception pretext data used in PPRL mainly comes from broadly available public resources (e.g., generic grounding/tracking and self-supervised counting). While this choice keeps the training pipeline lightweight and scalable, we do not systematically investigate whether higher-quality, artifact-specific grounding annotations could further improve detection performance and explanation fidelity. Exploring and curating such high-quality artifact grounding data, as well as integrating it into the pretext RL stage, is a promising direction for future work.

*Table 10.* Performance comparison on video datasets, including Accuracy (Acc), Recall (R) and F1. * means Average Precision (AP) is adopted following the official guideline. The final average performance is calculated by first averaging the results across ID (in-domain), OOD (out-of-domain) and OOD-MintVid, and then taking the average of the three values. The best results are **bolded**.

| Method | Metric | ID | OOD | | | | | | OOD-MintVid | | | | | | | | | | Avg. |
|---|---|---|---|---|---|---|---|---|---|---|---|---|---|---|---|---|---|---|---|
| | | | GenBuster++ | LOKI | Vidu Q1 | Gen-4 | Veo3 | Emu3 | Phantom-14B | OmniAvatar | FantasyTalking | Seedance1.0Pro | Jimeng3.0Pro | Kling2.5-Turbo | Hailuo2.3 | Wan2.5 | Sora2 | Fact | |
| *Small Vision Models* | | | | | | | | | | | | | | | | | | | |
| D3* (*ICCV'25*) | AP | 49.5 | 46.8 | 39.1 | 55.1 | 89.0 | 66.1 | 90.0 | 54.8 | 36.4 | 44.1 | 61.6 | 48.7 | 51.7 | 78.9 | 50.5 | 54.6 | 32.4 | 55.1 |
| DeMamba | Acc | 87.6 | 86.8 | 71.1 | 80.7 | 83.4 | 79.4 | 85.9 | 57.4 | 55.2 | 62.2 | 49.2 | 46.2 | 46.6 | 41.1 | 49.3 | 41.4 | 73.3 | 73.7 |
| | R | 87.2 | 90.8 | 84.4 | 77.7 | 81.5 | 77.7 | 87.8 | 50.4 | 58.1 | 60.0 | 72.4 | 57.6 | 58.3 | 54.1 | 64.1 | 48.1 | 73.8 | 76.7 |
| | F1 | 87.3 | 87.3 | 79.5 | 80.1 | 83.1 | 79.1 | 86.1 | 54.2 | 57.9 | 61.3 | 51.1 | 50.6 | 51.2 | 35.1 | 54.8 | 43.0 | 79.5 | 74.6 |
| NSG-VD (*NIPS'25*) | Acc | 52.4 | 53.3 | 55.1 | 53.4 | 58.2 | 64.3 | 50.7 | 54.3 | 53.8 | 58.5 | 53.8 | 58.0 | 56.0 | 69.4 | 57.0 | 57.8 | 52.8 | 55.1 |
| | R | 28.0 | 76.0 | 75.2 | 60.2 | 84.4 | 87.2 | 1.5 | 48.2 | **86.4** | 67.5 | 17.2 | 36.1 | 75.9 | 83.1 | 80.9 | **77.3** | 30.6 | 50.8 |
| | F1 | 27.9 | 61.9 | 62.5 | 56.4 | 66.9 | 71.0 | 3.0 | 51.4 | **65.1** | 61.9 | 27.1 | 46.3 | 63.3 | 73.2 | 65.3 | **64.8** | 39.3 | 45.8 |
| ReStraV (*NIPS'25*) | Acc | 50.7 | 48.9 | 64.8 | 54.8 | 52.8 | 59.8 | 95.2 | 49.6 | 49.3 | 49.9 | 36.9 | 48.4 | 47.2 | 30.2 | 40.9 | 45.4 | 65.0 | 53.2 |
| | R | 4.3 | 3.8 | 5.6 | 7.8 | 8.0 | 18.2 | 96.2 | 3.7 | 3.7 | 3.7 | 2.5 | 2.5 | 2.5 | 2.5 | 2.5 | 2.5 | 11.1 | 10.4 |
| | F1 | 8.0 | 6.9 | 9.7 | 14.0 | 14.5 | 30.0 | 95.3 | 6.8 | 6.8 | 6.8 | 4.9 | 4.9 | 4.8 | 4.9 | 4.8 | 4.8 | 16.0 | 14.3 |
| *Generic MLLMs* | | | | | | | | | | | | | | | | | | | |
| Qwen2.5-VL-7B | Acc | 54.1 | 59.5 | 49.4 | 58.8 | 53.3 | 60.3 | 99.2 | 50.7 | 50.0 | 49.7 | 63.6 | 49.4 | 51.6 | 53.1 | 55.8 | 50.4 | 87.1 | 57.9 |
| | R | 13.5 | 23.5 | 27.9 | 15.4 | 8.0 | 18.2 | 99.7 | 1.2 | 0.1 | 0.0 | 14.7 | 6.3 | 7.7 | 8.8 | 19.0 | 6.2 | 84.6 | 20.1 |
| | F1 | 21.7 | 36.2 | 42.2 | 26.1 | 14.7 | 30.2 | 99.3 | 2.3 | 0.3 | 0.0 | 23.9 | 11.3 | 13.7 | 15.6 | 30.7 | 11.2 | 87.8 | 27.6 |
| Qwen3-VL-8B | Acc | 65.1 | 62.4 | 56.6 | 60.2 | 59.4 | 62.9 | **99.9** | 53.3 | 50.6 | 50.3 | 64.2 | 51.8 | 55.1 | 54.1 | 66.7 | 50.8 | 94.6 | 65.5 |
| | R | 33.1 | 28.5 | 38.8 | 20.5 | 18.8 | 26.3 | **100.0** | 7.0 | 1.4 | 0.9 | 20.7 | 12.5 | 19.2 | 13.0 | 41.6 | 7.4 | 93.6 | 31.2 |
| | F1 | 44.3 | 43.1 | 54.2 | 34.0 | 31.6 | 41.5 | **99.9** | 13.0 | 2.8 | 1.7 | 31.8 | 21.1 | 30.7 | 21.8 | 56.4 | 13.0 | 92.6 | 41.2 |
| InternVL3.5-8B | Acc | 51.7 | 59.6 | 44.1 | 56.1 | 53.4 | 56.7 | 99.7 | 51.3 | 50.2 | 49.5 | 62.6 | 49.8 | 49.1 | 50.2 | 53.9 | 54.1 | 91.7 | 56.5 |
| | R | 9.3 | 8.6 | 14.2 | 9.7 | 6.3 | 5.9 | 99.7 | 1.3 | 1.9 | 0.9 | 5.9 | 3.2 | 4.1 | 2.2 | 7.8 | 5.3 | 85.1 | 15.0 |
| | F1 | 16.3 | 15.5 | 24.5 | 17.3 | 11.9 | 11.2 | 99.7 | 2.5 | 3.8 | 1.8 | 10.9 | 6.2 | 7.8 | 4.3 | 14.2 | 10.0 | 91.4 | 20.5 |
| MiMo-VL-7B | Acc | 65.8 | 54.3 | 53.6 | 65.5 | 65.8 | 66.0 | 98.8 | 58.4 | 50.1 | 50.6 | 69.7 | 56.7 | 61.7 | 56.1 | 66.0 | 55.4 | 89.9 | 64.8 |
| | R | 42.4 | 20.3 | 37.0 | 32.6 | 33.2 | 34.0 | **100.0** | 21.8 | 5.2 | 6.2 | 33.2 | 21.5 | 31.5 | 16.4 | 39.6 | 16.2 | 95.0 | 38.0 |
| | F1 | 52.7 | 30.7 | 51.4 | 48.5 | 49.3 | 50.0 | 98.8 | 34.4 | 9.5 | 11.2 | 46.9 | 33.9 | 46.0 | 27.0 | 54.7 | 26.5 | 91.1 | 48.5 |
| Keye-VL-1.5-8B | Acc | 52.8 | 62.4 | 60.4 | 58.3 | 56.0 | 62.4 | 94.7 | 51.3 | 51.1 | 53.3 | 59.7 | 50.5 | 51.0 | 52.8 | 52.9 | 51.8 | 80.4 | 58.0 |
| | R | 13.3 | 29.0 | 39.2 | 14.4 | 14.8 | 29.0 | 97.7 | 3.9 | 4.6 | 7.4 | 23.9 | 13.1 | 13.4 | 13.5 | 18.1 | 16.0 | 84.0 | 23.5 |
| | F1 | 19.9 | 43.8 | 50.4 | 24.7 | 24.6 | 43.8 | 94.0 | 7.3 | 8.7 | 13.6 | 34.7 | 21.2 | 21.5 | 22.2 | 28.0 | 25.2 | 81.3 | 31.1 |
| GLM-4.5V | Acc | 61.7 | 59.4 | 49.4 | 56.5 | 56.6 | 62.4 | 97.1 | 57.9 | 52.2 | 50.5 | 63.2 | 48.9 | 49.2 | 49.8 | 58.3 | 47.9 | 88.2 | 60.6 |
| | R | 30.4 | 27.7 | 32.0 | 26.6 | 17.3 | 29.8 | 99.2 | 18.4 | 6.5 | 2.6 | 27.4 | 16.6 | 15.7 | 13.8 | 32.7 | 11.9 | 95.2 | 31.1 |
| | F1 | 43.3 | 40.6 | 46.4 | 40.4 | 28.5 | 44.2 | 97.1 | 30.3 | 12.0 | 5.0 | 39.2 | 26.8 | 25.5 | 22.9 | 46.7 | 19.9 | 90.1 | 41.6 |
| Qwen3-VL-235B-A22B | Acc | 66.7 | 64.6 | 57.9 | 57.4 | 65.7 | 63.4 | 98.5 | 54.1 | 50.9 | 51.2 | 65.8 | 49.1 | 32.3 | 49.8 | 63.5 | 50.5 | 93.6 | 63.6 |
| | R | 38.7 | 36.5 | 44.7 | 29.3 | 31.9 | 28.7 | **100.0** | 9.1 | 2.6 | 3.3 | 32.3 | 16.2 | 32.3 | 13.7 | 41.3 | 16.2 | 94.3 | 36.7 |
| | F1 | 51.6 | 50.8 | 59.4 | 45.0 | 48.1 | 43.8 | 98.5 | 16.5 | 4.9 | 6.3 | 45.1 | 26.4 | 48.8 | 22.7 | 55.8 | 26.2 | 94.3 | 48.0 |
| Gemini-2.5-Pro | Acc | 80.5 | 71.8 | 70.2 | 70.8 | 76.0 | 70.4 | 93.3 | 57.8 | 55.8 | 55.6 | 69.4 | 60.2 | 65.6 | 62.7 | 65.7 | 60.9 | 88.3 | 73.4 |
| | R | 85.1 | 59.8 | 65.5 | 38.2 | 65.6 | 46.8 | **100.0** | 28.2 | 22.6 | 18.2 | 52.0 | 34.9 | 47.7 | 39.4 | 47.7 | 39.4 | 95.5 | 63.4 |
| | F1 | 81.4 | 68.1 | 70.3 | 55.3 | 74.7 | 60.4 | 93.7 | 40.6 | 33.8 | 27.9 | 60.9 | 45.9 | 58.5 | 51.0 | 58.4 | 51.0 | 90.3 | 67.9 |
| Gemini-3-Pro-Preview | Acc | 79.4 | 82.5 | 74.9 | 83.1 | 76.3 | 85.7 | 89.3 | 65.5 | **65.3** | 61.5 | 81.4 | 72.1 | 73.1 | 74.3 | 73.1 | 63.5 | 91.4 | 77.8 |
| | R | 96.2 | 93.5 | **94.4** | **92.7** | 75.8 | 89.6 | **100.0** | 60.8 | 60.2 | 52.7 | **81.2** | **71.8** | 73.9 | 76.0 | 73.9 | 54.4 | 98.6 | 86.3 |
| | F1 | 82.7 | 84.3 | 80.2 | 84.7 | 76.1 | 86.3 | 90.3 | 63.9 | 64.3 | 57.6 | 81.3 | 71.9 | 73.1 | 74.5 | 73.3 | 59.7 | 92.6 | 79.1 |
| *MLLM-based Video Forgery Detectors* | | | | | | | | | | | | | | | | | | | |
| BusterX++ | Acc | 77.1 | 79.6 | 70.9 | 53.0 | 89.9 | 76.2 | 98.9 | 62.3 | 48.6 | 47.6 | 71.8 | 71.7 | 82.3 | 70.4 | **87.3** | 58.7 | 88.8 | 74.7 |
| | R | 64.9 | 68.8 | 67.5 | 6.1 | 82.9 | 55.5 | **100.0** | 30.6 | 3.3 | 1.3 | 47.8 | 47.3 | 73.9 | 45.6 | **86.2** | 13.0 | 90.0 | 57.4 |
| | F1 | 71.4 | 77.1 | 75.5 | 11.4 | 89.2 | 70.0 | 98.9 | 44.8 | 6.0 | 2.4 | 57.7 | 57.2 | 77.1 | 55.6 | 84.5 | 19.8 | 89.7 | 63.7 |
| Skyra-RL | Acc | 52.1 | 55.7 | 37.7 | 52.9 | 59.1 | 55.9 | 51.5 | 50.1 | 49.9 | 49.9 | 63.4 | 51.7 | 53.5 | 51.7 | 59.6 | 50.6 | 51.9 | 52.5 |
| | R | 4.6 | 11.7 | 7.3 | 6.1 | 18.2 | 12.1 | 3.0 | 0.3 | 0.0 | 0.0 | 9.9 | 6.9 | 10.8 | 2.7 | 22.3 | 1.0 | 11.4 | 6.9 |
| | F1 | 8.4 | 20.9 | 13.5 | 11.4 | 30.7 | 21.5 | 5.8 | 0.6 | 0.0 | 0.0 | 17.9 | 12.8 | 19.4 | 5.3 | 36.4 | 2.0 | 20.5 | 12.4 |
| **VIDEOVERITAS (ours)** | Acc | **93.1** | **91.4** | 78.1 | **93.6** | **96.7** | **94.5** | 99.4 | **79.0** | 56.2 | **84.5** | **86.6** | **80.8** | **86.0** | 85.8 | 86.3 | 67.6 | **96.1** | **88.8** |
| | R | **99.3** | **94.9** | 91.0 | 89.5 | 96.0 | 92.3 | **100.0** | 76.6 | 30.5 | **87.5** | 75.7 | 68.4 | **78.6** | 78.1 | 79.3 | 41.1 | **100.0** | **88.3** |
| | F1 | **93.5** | **91.7** | 84.7 | **93.3** | **96.7** | **94.3** | 99.4 | **78.5** | 40.9 | **84.9** | **82.0** | **78.6** | **85.4** | **84.4** | **85.8** | 55.8 | **96.6** | **88.0** |

*Table 11.* Detailed ablation of perception tasks on MindVid dataset. The reported metrics are "Acc/Recall/F1".

| Task | Phantom-14B | OmniAvatar | FantasyTalking | Seedance1.0Pro | Jimeng3.0Pro | Kling2.5-Turbo | Hailuo2.3 | Wan2.5 | Sora2 |
|---|---|---|---|---|---|---|---|---|---|
| w/o Perception | 77.1/76.0/76.8 | 52.6/27.1/36.4 | 84.7/91.2/85.6 | 81.3/67.5/74.5 | 76.2/62.6/73.1 | 83.5/76.9/82.9 | 82.8/75.3/81.2 | 83.0/75.9/82.2 | 61.1/31.3/44.5 |
| SSL | **81.0/86.1/81.9** | **62.5/49.1/56.7** | 85.5/**95.2**/86.8 | 84.1/74.3/78.9 | 78.9/67.9/76.9 | 83.7/77.2/83.1 | 82.4/78.1/81.4 | 84.6/78.9/84.1 | 65.5/40.1/53.6 |
| G-G | 76.2/72.3/75.3 | 58.2/36.2/46.4 | 84.2/88.2/84.8 | 85.3/76.4/80.7 | 79.4/68.2/77.4 | 85.1/79.3/84.7 | **86.1/82.9/85.5** | 83.9/77.1/83.3 | 66.5/41.4/55.2 |
| A-G | 79.5/79.7/79.5 | 58.2/37.2/47.1 | 85.2/91.2/86.1 | 84.5/73.9/79.4 | 79.4/67.8/77.2 | 84.3/77.5/83.7 | **86.1**/82.2/85.4 | 84.4/77.6/83.8 | **68.7/45.5/59.1** |
| SSL + G-G | 79.0/76.6/78.5 | 56.0/30.5/40.9 | 84.5/87.5/84.9 | **86.6**/75.7/82.0 | **80.8**/68.4/**78.6** | **86.0**/78.6/**85.4** | 85.8/78.1/84.4 | **86.3/79.3/85.8** | 67.6/41.1/55.8 |
| SSL + A-G | 79.8/80.6/79.9 | 58.8/38.6/48.4 | 86.2/93.3/87.1 | 82.7/70.0/76.5 | 78.1/65.7/75.6 | 83.8/76.9/83.1 | 83.8/78.1/82.6 | 83.6/76.5/82.9 | 64.2/36.7/50.5 |
| G-G + A-G | 78.1/80.4/78.6 | 57.1/38.4/47.3 | 84.9/94.0/86.2 | 86.4/**79.6**/**82.5** | 79.7/**69.2**/77.9 | 85.1/**79.6**/84.7 | 85.1/81.5/84.4 | 84.8/78.9/84.3 | 63.7/36.0/49.7 |
| SSL + G-G + A-G | 79.1/76.9/78.7 | 60.1/38.9/49.4 | **87.8/94.2/88.5** | 81.3/68.5/74.7 | 74.4/59.8/70.7 | 81.9/74.4/81.0 | 81.1/75.3/79.7 | 83.3/77.1/82.7 | 64.2/38.0/51.4 |

**System Prompt for Detection**

You are an expert video analyst.
Please think about the question as if you were a human pondering deeply. It's encouraged to include self-reflection or verification in the reasoning process. Then, give a final verdict within `<answer> </answer>` tags.

*Figure 12.* System prompt for detection task.

**User Prompt for Detection**

```
<video>
```
Is this video real or fake?

*Figure 13.* User prompt for detection task.

**System Prompt for Perception**

You are an expert video analyst. Based on this video, provide the answer directly.

*Figure 14.* System prompt for perception task.

**User Prompt for Spatiotemporal Grounding**

```
<video>
```
Given the query "`[description]`", when and where does the described content occur in the video? please firstly give the start and end time, spatial bounding box corresponding to each integer second.

Please provide only the time span in seconds and bounding boxes as JSON, ONLY up to 16 seconds.
You MUST output one bounding box for every integer second within the given time span (inclusive).
Example:
{"time": [8.125, 13.483], "boxes": {"9": [317, 422, 582, 997], "10": [332, 175, 442, 369], "11": [340, 180, 450, 370]}}
Note: Each key in "boxes" must be an integer second within the span, and its value must be a 4-number bounding box [x1, y1, x2, y2].

*Figure 15.* User prompt for spatiotemporal grounding task. "`[description]`" is a brief caption of the target object, which is included in the dataset, such as "a baby closes a laptop in the study".

**User Prompt for Object Tracking**

```
<video>
```
Given the bounding box "`[initial box]`" of the target object in the first frame, track this object in each frame and output its bounding box once per second.ONLY up to 16 seconds.
Example:
{"boxes": {"1": [405, 230, 654, 463], "2": [435, 223, 678, 446], ..., "16": [415, 203, 691, 487]}}
Note: Each key in "boxes" must correspond to a second (1, 2, 3, ..., 16) and contain a 4-number bounding box [x1, y1, x2, y2].

*Figure 16.* User prompt for object tracking task. "`[initial box]`" is the bounding box in the first frame, which is included in the dataset.

---

**User Prompt for Object Counting**

```
<video>
```
Count the number of circles, squares, and triangles that appear in this video. Be aware that the shapes can appear in any color and at any angle of rotation. They may be present on one or multiple frames, and any given frame can contain more than one shape. Provide the answer as three comma-separated numbers in the format: circles,squares,triangles. For example, if you see 3 circles, 1 square, and 4 triangles, your answer should be "3,1,4".

*Figure 17.* User prompt for object counting task.

---

**User Prompt for Artifact Grounding**

```
<video>
```
Find the visual artifacts at "`[time]`" in the video.
Provide the bounding boxes where the artifact occurred, in [$x_{min}$, $y_{min}$, $x_{max}$, $y_{max}$] format. If there are multiple locations, you should provide all the bounding boxes.
Example:
{"boxes": [[487, 324, 573, 398], [670, 533, 734, 769], ...]]}.

*Figure 18.* User prompt for artifact grounding task (as mentioned in Sec. 4.3). "`[time]`" is the exact target timestamp in seconds, which is included in the dataset, such as "4.5s".

---

**Input Prompt for Reasoning Behavior Evaluation**

You are a helpful assistant proficient in analyzing vision reasoning problems.
## Instruction:
You will be presented with two analytical reports (Assistant A and Assistant B) that describe observations from a video. Your task is to perform a side-by-side comparison and determine which assistant provides higher-quality reasoning based **ONLY** on the specific dimension provided below.
The evaluation must be conducted strictly based on the textual evidence provided. Do not assume any external video information. Your goal is to identify which assistant demonstrates more professional, precise, and logically structured analysis within the specified scope.

## Evaluation Dimension: {`Dimension Name`}
**Description**: {`Dimension Description`}

## Rules for Evaluation:
- **Strictly Dimension-Focused**: Ignore other aspects of the reports. Only judge based on the provided dimension.
- **Content over Conclusion**: Do not favor an assistant based on its final verdict. Focus on the depth and quality of the reasoning path.
- **Neutrality**: Be unbiased toward length or tone. Prioritize the density of meaningful, analytical information.

## Desired Output Format:
Present your verdict in a JSON format, with key "analysis" for a short reason of your judgment and key "judgment" to indicate your decision: use "[[A]]" if assistant A prevails, "[[B]]" if assistant B does, and "[[C]]" for a tie.

## Input Data:
[The Start of Assistant A's Analysis]
{`Output of Assistant A`}
[The End of Assistant A's Analysis]
[The Start of Assistant B's Analysis]
{`Output of Assistant B`}
[The End of Assistant B's Analysis]

*Figure 19.* Input Prompt for Reasoning Behavior Evaluation (Figure 2).

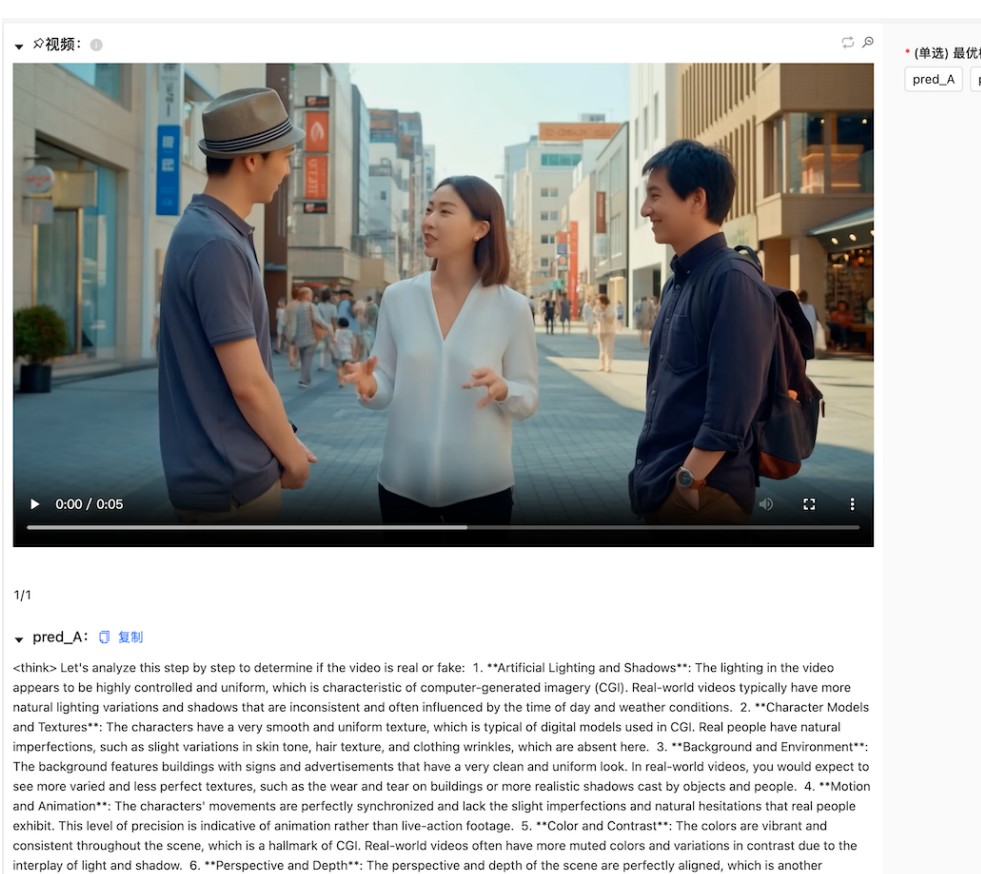

*Figure 20.* Human-judgment platform for pairwise reasoning-quality comparison.

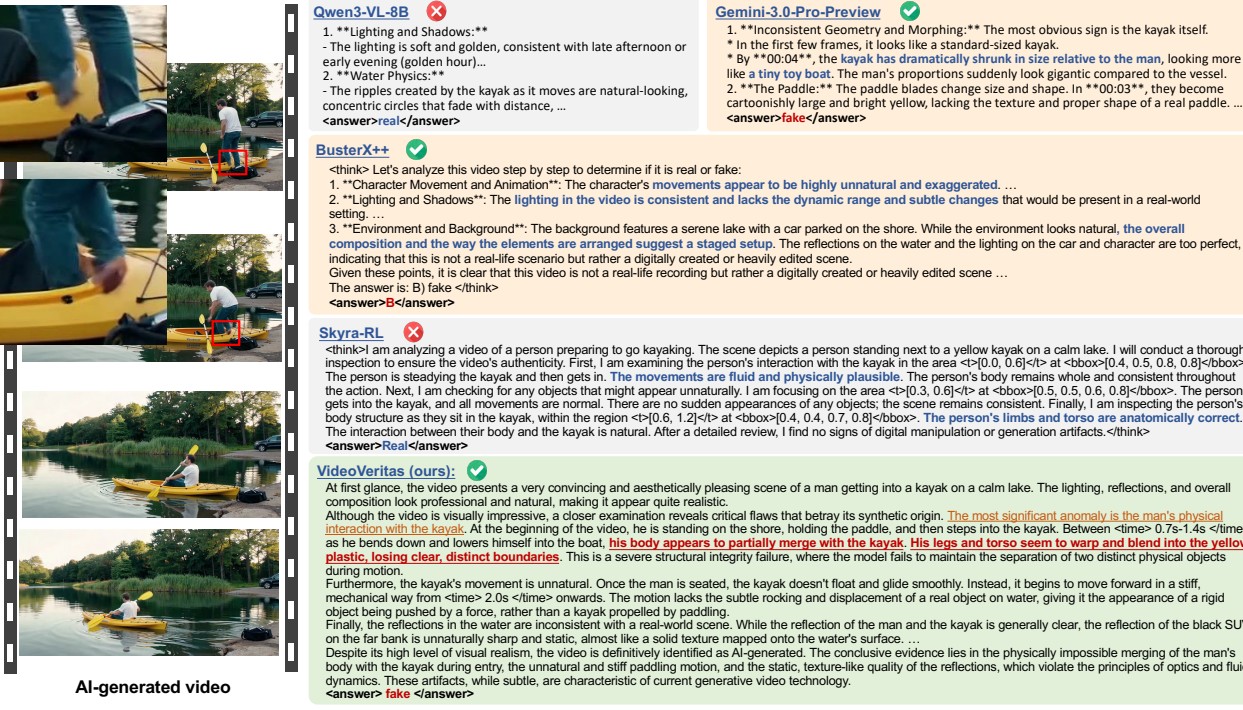

**AI-generated video**

**Qwen3-VL-8B** ❌
1. **Lighting and Shadows:**
- The lighting is soft and golden, consistent with late afternoon or early evening (golden hour)...
2. **Water Physics:**
- The ripples created by the kayak as it moves are natural-looking, concentric circles that fade with distance, ...
<answer>real</answer>

**Gemini-3.0-Pro-Preview** ✅
1. **Inconsistent Geometry and Morphing:** The most obvious sign is the kayak itself.
* In the first few frames, it looks like a standard-sized kayak.
* By **00:04**, the kayak has dramatically shrunk in size relative to the man, looking more like a tiny toy boat. The man's proportions suddenly look gigantic compared to the vessel.
2. **The Paddle:** The paddle blades change size and shape. In **00:03**, they become cartoonishly large and bright yellow, lacking the texture and proper shape of a real paddle. ...
<answer>fake</answer>

**BusterX++** ✅
<think> Let's analyze this video step by step to determine if it is real or fake:
1. **Character Movement and Animation**: The character's movements appear to be highly unnatural and exaggerated. …
2. **Lighting and Shadows**: The lighting in the video is consistent and lacks the dynamic range and subtle changes that would be present in a real-world setting. …
3. **Environment and Background**: The background features a serene lake with a car parked on the shore. While the environment looks natural, the overall composition and the way the elements are arranged suggest a staged setup. The reflections on the water and the lighting on the car and character are too perfect, indicating that this is not a real-life scenario but rather a digitally created or heavily edited scene.
Given these points, it is clear that this video is not a real-life recording but rather a digitally created or heavily edited scene …
The answer is: B) fake </think>
<answer>B</answer>

**Skyra-RL** ❌
<think> I am analyzing a video of a person preparing to go kayaking. The scene depicts a person standing next to a yellow kayak on a calm lake. I will conduct a thorough inspection to ensure the video's authenticity. First, I am examining the person's interaction with the kayak in the area <t>[0.0, 0.6]</t> at <bbox>[0.4, 0.5, 0.8, 0.8]</bbox>. The person is steadying the kayak and then gets in. The movements are fluid and physically plausible. The person's body remains whole and consistent throughout the action. Next, I am checking for any objects that might appear unnaturally. I am focusing on the area <t>[0.3, 0.6]</t> at <bbox>[0.5, 0.5, 0.6, 0.8]</bbox>. The person gets into the kayak, and all movements are normal. There are no sudden appearances of any objects; the scene remains consistent. Finally, I am inspecting the person's body structure as they sit in the kayak, within the region <t>[0.6, 1.2]</t> at <bbox>[0.4, 0.4, 0.7, 0.8]</bbox>. The person's limbs and torso are anatomically correct. The interaction between their body and the kayak is natural. After a detailed review, I find no signs of digital manipulation or generation artifacts.</think>
<answer>Real</answer>

**VideoVeritas (ours):** ✅
At first glance, the video presents a very convincing and aesthetically pleasing scene of a man getting into a kayak on a calm lake. The lighting, reflections, and overall composition look professional and natural, making it appear quite realistic.
Although the video is visually impressive, a closer examination reveals critical flaws that betray its synthetic origin. The most significant anomaly is the man's physical interaction with the kayak. At the beginning of the video, he is standing on the shore, holding the paddle, and then steps into the kayak. Between <time> 0.7s-1.4s </time>, as he bends down and lowers himself into the boat, his body appears to partially merge with the kayak. His legs and torso seem to warp and blend into the yellow plastic, losing clear, distinct boundaries. This is a severe structural integrity failure, where the model fails to maintain the separation of two distinct physical objects during motion.
Furthermore, the kayak's movement is unnatural. Once the man is seated, the kayak doesn't float and glide smoothly. Instead, it begins to move forward in a stiff, mechanical way from <time> 2.0s </time> onwards. The motion lacks the subtle rocking and displacement of a real object on water, giving it the appearance of a rigid object being pushed by a force, rather than a kayak propelled by paddling.
Finally, the reflections in the water are inconsistent with a real-world scene. While the reflection of the man and the kayak is generally clear, the reflection of the black SUV on the far bank is unnaturally sharp and static, almost like a solid texture mapped onto the water's surface. …
Despite its high level of visual realism, the video is definitively identified as AI-generated. The conclusive evidence lies in the physically impossible merging of the man's body with the kayak during entry, the unnatural and stiff paddling motion, and the static, texture-like quality of the reflections, which violate the principles of optics and fluid dynamics. These artifacts, while subtle, are characteristic of current generative video technology.
<answer> fake </answer>

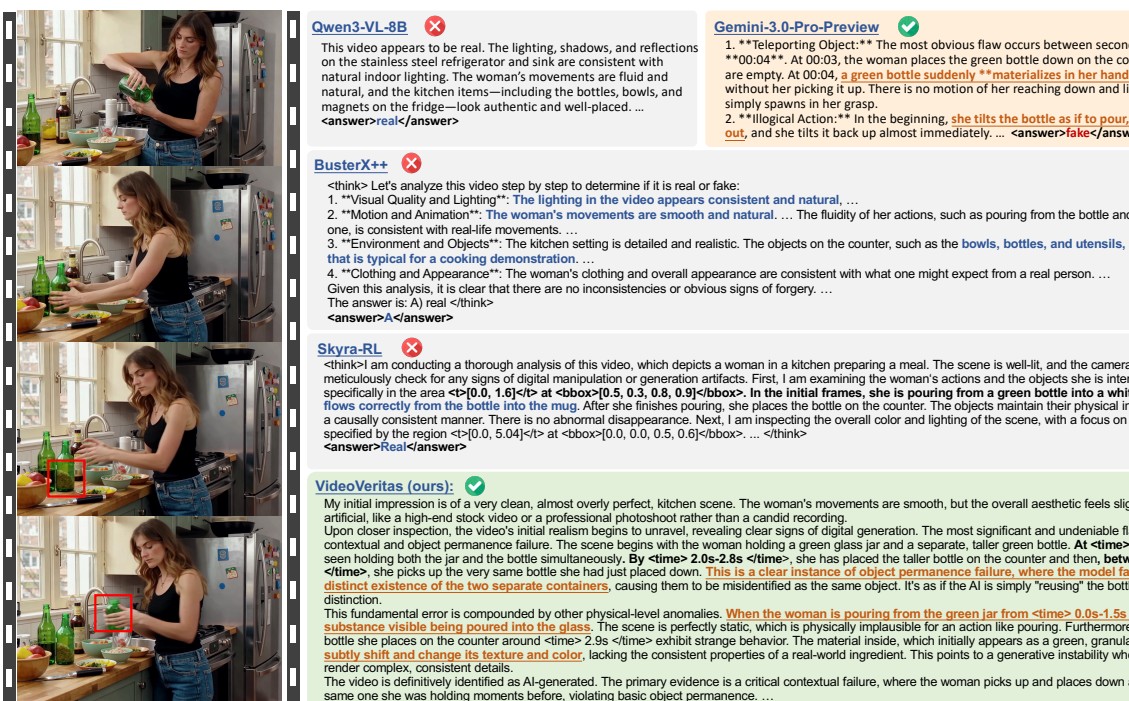

**AI-generated video**

**Qwen3-VL-8B** ❌
This video appears to be real. The lighting, shadows, and reflections on the stainless steel refrigerator and sink are consistent with natural indoor lighting. The woman's movements are fluid and natural, and the kitchen items—including the bottles, bowls, and magnets on the fridge—look authentic and well-placed. …
<answer>real</answer>

**Gemini-3.0-Pro-Preview** ✅
1. **Teleporting Object:** The most obvious flaw occurs between seconds **00:03** and **00:04**. At 00:03, the woman places the green bottle down on the counter and her hands are empty. At 00:04, a green bottle suddenly **materializes in her hands** at chest height without her picking it up. There is no motion of her reaching down and lifting it back up; it simply spawns in her grasp.
2. **Illogical Action:** In the beginning, she tilts the bottle as if to pour, but no liquid comes out, and she tilts it back up almost immediately. … <answer>fake</answer>

**BusterX++** ❌
<think> Let's analyze this video step by step to determine if it is real or fake:
1. **Visual Quality and Lighting**: The lighting in the video appears consistent and natural, …
2. **Motion and Animation**: The woman's movements are smooth and natural. … The fluidity of her actions, such as pouring from the bottle and reaching for another one, is consistent with real-life movements. …
3. **Environment and Objects**: The kitchen setting is detailed and realistic. The objects on the counter, such as bowls, bottles, and utensils, are placed in a way that is typical for a cooking demonstration. …
4. **Clothing and Appearance**: The woman's clothing and overall appearance are consistent with what one might expect from a real person. …
Given this analysis, it is clear that there are no inconsistencies or obvious signs of forgery. …
The answer is: A) real </think>
<answer>A</answer>

**Skyra-RL** ❌
<think> I am conducting a thorough analysis of this video, which depicts a woman in a kitchen preparing a meal. The scene is well-lit, and the camera is stable. I will meticulously check for any signs of digital manipulation or generation artifacts. First, I am examining the woman's actions and the objects she is interacting with, specifically in the area <t>[0.0, 1.6]</t> at <bbox>[0.5, 0.3, 0.8, 0.9]</bbox>. In the initial frames, she is pouring from a green bottle into a white mug. The liquid flows correctly from the bottle into the mug. After she finishes pouring, she places the bottle on the counter. The objects maintain their physical integrity and interact in a causally consistent manner. There is no abnormal disappearance. Next, I am inspecting the overall color and lighting of the scene, with a focus on the background, as specified by the region <t>[0.0, 5.04]</t> at <bbox>[0.0, 0.0, 0.5, 0.6]</bbox>. … </think>
<answer>Real</answer>

**VideoVeritas (ours):** ✅
My initial impression is of a very clean, almost overly perfect, kitchen scene. The woman's movements are smooth, but the overall aesthetic feels slightly sterile and artificial, like a high-end stock video or a professional photoshoot rather than a candid recording.
Upon closer inspection, the video's initial realism begins to unravel, revealing clear signs of digital generation. The most significant and undeniable flaw is a severe contextual and object permanence failure. The scene begins with the woman holding a green glass jar and a separate, taller, green bottle. At <time> 1.5s </time>, she is seen holding both the jar and the bottle simultaneously. By <time> 2.0s-2.8s </time>, she has placed the taller bottle on the counter and then, between <time> 2.9s-3.2s </time>, she picks up the very same bottle she had just placed down. This is a clear instance of object permanence failure, where the model fails to maintain the distinct existence of the two separate containers, causing them to be misidentified as the same object. It's as if the AI is simply "reusing" the bottle asset without proper distinction.
This fundamental error is compounded by other physical-level anomalies. When the woman is pouring from the green jar from <time> 0.0s-1.5s </time>, there is no substance visible being poured into the glass. The scene is perfectly static, which is physically implausible for an action like pouring. Furthermore, the contents of the bottle she places on the counter around <time> 2.9s </time> exhibit strange behavior. The material inside, which initially appears as a green, granular substance, seems to subtly shift and change its texture and color, lacking the consistent properties of a real-world ingredient. This points to a generative instability where the AI struggles to render complex, consistent details.
The video is definitively identified as AI-generated. The primary evidence is a critical contextual failure, where the woman picks up and places down a bottle that is the same one she was holding moments before, violating basic object permanence. …
<answer> fake </answer>

*Figure 21.* Reasoning comparisons between VIDEOVERITAS and existing MLLM-based detectors.

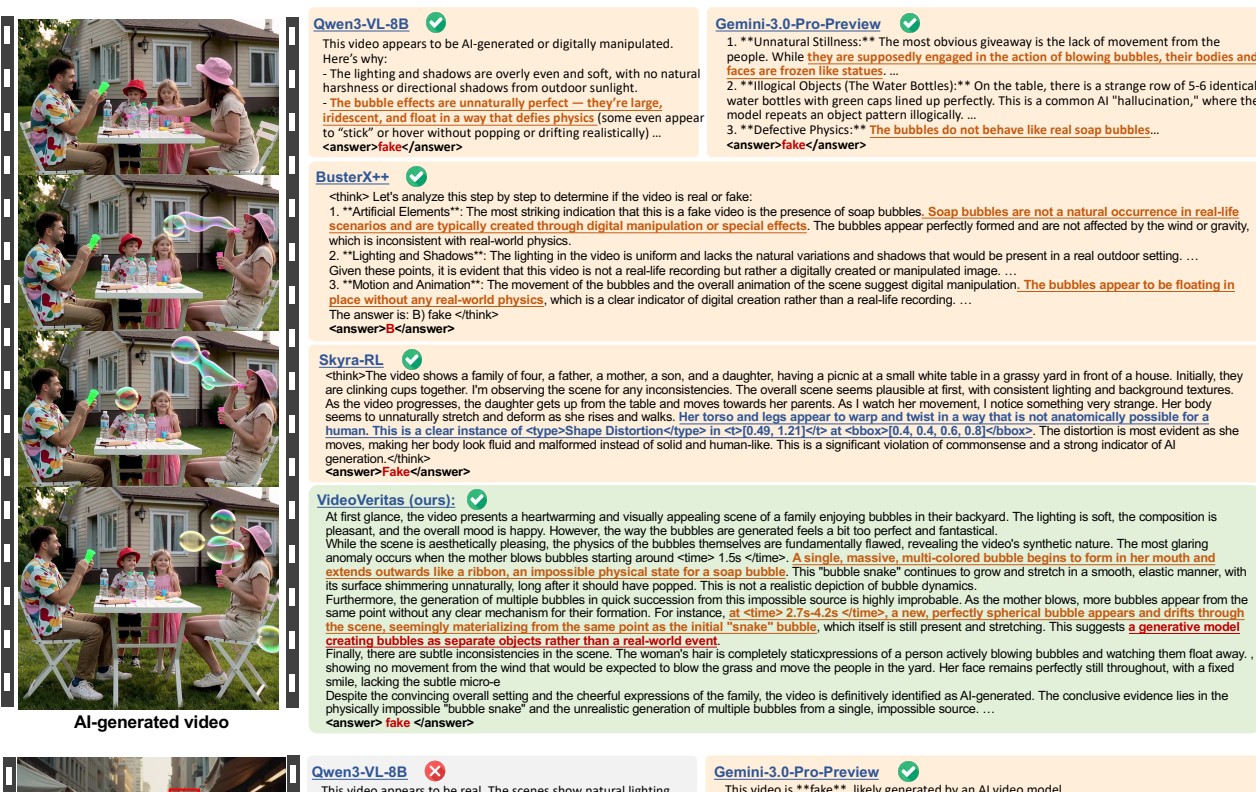

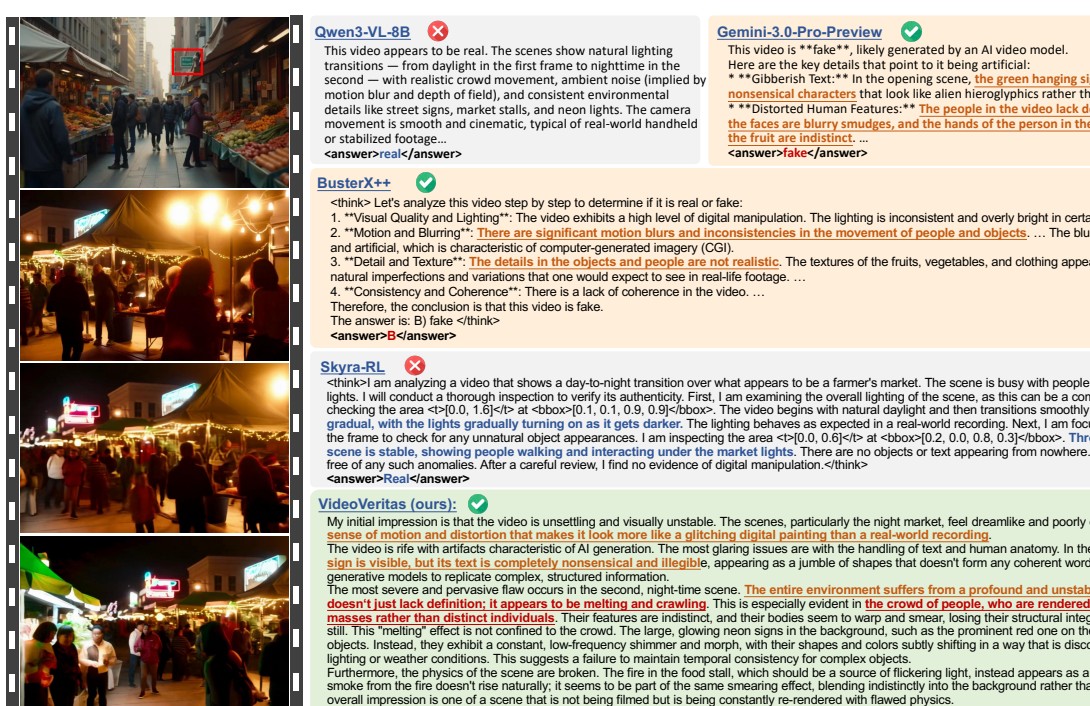

*Figure 22.* Reasoning comparisons between **VIDEOVERITAS** and existing MLLM-based detectors.

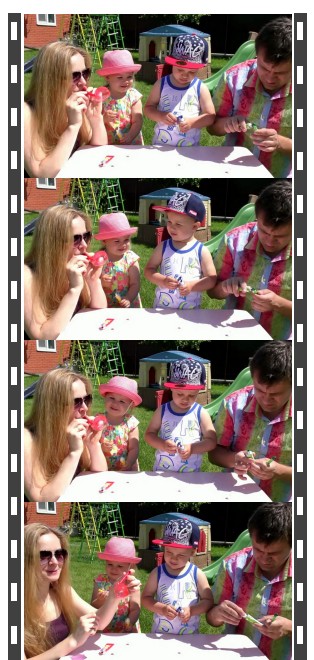

**Qwen3-VL-8B** ✅
This video appears to be real. The lighting, shadows, and natural movements of the people (especially the children's expressions and the way they interact with the objects) look authentic. The background elements — like the playground equipment, the house, and the grass — are consistent with a real outdoor setting. There are no obvious signs of CGI, editing artifacts, or unnatural motion that would suggest it's fake.
<answer>real</answer>

**Gemini-3.0-Pro-Preview** ✅
The video appears to be a genuine home movie. Here is the reasoning:
1. **Coherent Text and Patterns:** One of the biggest tells in AI-generated video is garbled text. The boy's hat clearly reads "POLICE," and the letters are stable and legible throughout the frames. ...
2. **Natural Physics and Lighting:** The scene is lit by harsh sunlight, creating distinct, hard shadows (e.g., under the boy's hat brim and the woman's chin). ...
3. **Anatomical Accuracy:** The hands and fingers of all subjects look anatomically correct. ...
<answer>real</answer>

**BusterX++** ✅
<think> Let's analyze this video step by step to determine if it is real or fake:
1. **Setting and Environment**: The video shows a family outdoors in a backyard setting. There are elements like a playhouse, a slide, and a swing set, which are typical of a residential backyard. The lighting and shadows suggest natural sunlight, indicating it is daytime. The grass appears to be well-maintained, and the overall environment looks like a real, lived-in space.
2. **Characters and Actions**: The individuals in the video appear to be a family consisting of two adults and two children. ... The expressions and body language of the individuals are consistent with real human behavior, ...
3. **Clothing and Accessories**: The clothing and accessories worn by the individuals are casual and appropriate for a family outing in warm weather. ...
Therefore, the conclusion is that this video is real.
The answer is: A) real </think>
<answer>A</answer>

**Skyra-RL** ✅
<think>I am analyzing a video of a family of four at a table outdoors. The scene appears to be a sunny day in a backyard. I will conduct a thorough inspection to ensure the video's authenticity, paying close attention to common areas of digital manipulation. My first point of inspection is the man on the right, as clothing textures can often reveal artifacts. I am examining his colorful shirt in the area <t>[0, 0.68]</t> at <bbox>[0.7, 0.4, 1.0, 0.9]</bbox>. The pattern on his shirt is stable and clear. The individual squares and shapes that make up the design are well-defined and do not exhibit any unnatural jittering, flickering, or crawling. The texture is consistent and looks natural. Having found no evidence of forgery, I conclude the video is authentic.</think>
<answer>Real</answer>

**VideoVeritas (ours):** ✅
My initial impression is that this is a genuine, candid home video of a family playing with bubble solution in their backyard. The lighting, the children's behavior, and the overall scene feel completely natural and unscripted.
The video's authenticity is strongly supported by the complex and natural human behaviors on display. The woman on the left is blowing a bubble, and the process is not perfectly smooth. Around <time> 3.0s </time>, the bubble pops, and she reacts with a natural, slightly surprised expression, which is a common and unscripted human response. Her subsequent smile and gesture towards the bubble are also spontaneous and consistent with the playful context.
The children's actions further reinforce the video's realism. The boy in the center is intently focused on his small tube of bubble solution, his brow furrowed in concentration as he tries to open it. This is a typical, nuanced expression of a young child's focused effort. The girl in the pink hat watches the woman with a calm, observant expression, her posture and gaze natural for a toddler. These are not the stiff or overly simplistic movements often seen in generated videos.
Furthermore, the physical environment is rendered with complete consistency. The scene is lit by a strong, direct sun, casting sharp, well-defined shadows on the white table and the ground. These shadows move and deform correctly as the people shift their positions. For instance, the shadow of the man's hands and the green tube he is holding is cast realistically onto the table. The textures of the clothing, the grass, and the plastic toys are stable and detailed, with no signs of the flickering or "crawling" artifacts that often betray AI generation.
The video is conclusively real. The evidence lies in the combination of natural, unscripted human behaviors, such as the woman's reaction to the popped bubble and the children's focused concentration, and the physically consistent environment, including stable textures and accurate lighting and shadows. These elements demonstrate a level of organic complexity and imperfection that is characteristic of a genuine recording.
<answer> real </answer>

**Realistic video**

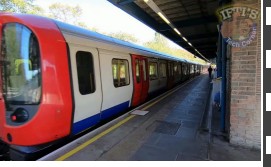
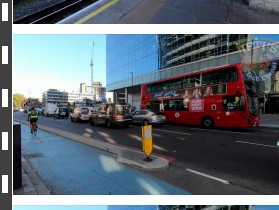
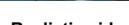

**Qwen3-VL-8B** ✅
**1. Scene 1: Train at a station**
- The train has a red, white, and blue livery — very characteristic of London Underground trains (specifically, the District Line or similar)...
**2. Scene 2: City street with traffic**
- The architecture includes modern glass buildings, a construction crane, and a red double-decker bus — all very typical of London.
- The bus has an advertisement for "Roast Battle" — a real TV show that aired on ITV, and the ad placement and design look authentic.
<answer>real</answer>

**Gemini-3.0-Pro-Preview** ✅
1. **Optical Characteristics**: The footage exhibits distinct characteristics of a wide-angle lens, likely from an action camera (like a GoPro or Insta360) or a 360-degree camera reframed. ...
2. **Scene Details and Consistency**:
* **Text & Branding**: In the street scene (Frames 3-5), the text on the vehicles is legible and consistent. The black van clearly says "Taskrabbit," and the police van has correct "POLICE" livery. AI video generators currently struggle significantly with maintaining clear, readable, and temporally consistent text.
* **Geographical Accuracy**: The visual elements are perfectly consistent with London, UK. Frames 1-2 show a London Underground Central Line train (1992 Stock) at an above-ground station. ...
<answer>real</answer>

**BusterX++** ✅
<think> Let's analyze this video step by step to determine if it is real or fake:
1. **Train Station Scene**:
- The first part of the video shows a train station with a train pulling out. The environment looks like a typical urban train station with tracks, platforms, and a station building. The lighting, shadows, and reflections on the train and platform are consistent with natural daylight. ...
2. **Urban Street Scene**:
- The second part of the video transitions to an urban street scene. The presence of vehicles, including a red double-decker bus, a police van, and various cars, is typical of a busy city environment. The bus has advertisements that are clearly visible and realistic, indicating it's a real bus in a real city...
The answer is: A) real </think>
<answer>A</answer>

**Skyra-RL** ✅
<think>I am analyzing a video clip that starts with a shot of a train at a station and then transitions to a city street scene. I will conduct a thorough inspection to verify its authenticity. My primary focus will be on areas where generation artifacts commonly appear, such as complex textures and motion. I am specifically examining the transition between the two scenes, as this can be a point of failure. I am checking the area <t>[0.4, 0.9]</t> at <bbox>[0.0, 0.3, 0.5, 0.9]</bbox>. In this video, the transition is a simple, clean cut from the train shot to the street scene. Both camera movements (the static shot of the train and the subsequent pan of the street) are smooth and consistent. The motion is fluid and there is no evidence of unnatural jittering, warping, or texture degradation. The overall visual quality is consistent and natural. Having found no signs of digital manipulation, I conclude the video is authentic.</think>
<answer>Real</answer>

**VideoVeritas (ours):** ✅
My initial impression is that this is a genuine video, likely a compilation of two separate clips. The scenes depict a London train station and a busy city street, both of which are rich with authentic, complex details that would be difficult to generate synthetically.
The video's authenticity is strongly supported by the high level of physical and contextual detail present in both scenes. The first segment, showing a London Underground train at a station, is filled with realistic elements. The lighting is consistent, with bright, natural sunlight casting plausible shadows on the platform and the train. The reflections on the train's curved windows are complex and accurate, mirroring the trees and sky in a way that is characteristic of real-world optics.
The second segment, a street scene, is even more convincing. It features a classic London red double-decker bus with a detailed, multi-layered advertisement for "Roast Battle" that is perfectly legible. The scene is populated with a variety of vehicles, including a black van with a "Passionate about our coffee" ad, a police van, and a silver car, all moving in a natural, unscripted manner. The presence of two cyclists in bright, high-visibility gear is also a common and logical sight in a busy urban environment. The reflections on the glass facade of the modern building are sharp and correspond correctly to the surrounding cityscape and sky. The entire scene is a complex, coherent snapshot of a real place and time, with no signs of the structural or semantic inconsistencies often seen in AI-generated videos.
The video is conclusively real. The evidence lies in the accumulation of authentic, specific, and complex details that are hallmarks of a genuine recording. From the correct train livery and destination to the detailed, legible advertisements on the bus and the natural, unscripted flow of traffic, every element points to a real-world location. The physical properties, such as lighting, shadows, and reflections, are all consistent and realistic, confirming the video's authenticity.
<answer> real </answer>

**Realistic video**

*Figure 23.* Reasoning comparisons between **VIDEOVERITAS** and existing MLLM-based detectors.

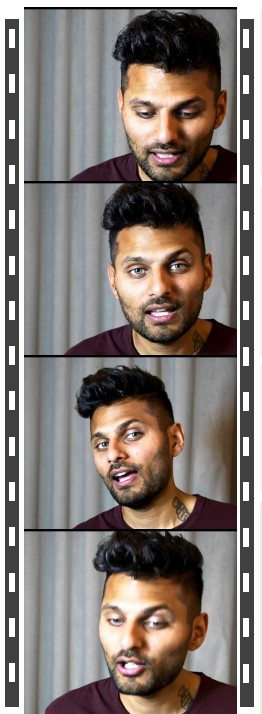

**Ai-generated video, facial**

**Qwen3-VL-8B** ❌
1. **Visual Consistency**: The man's facial expressions, mouth movements, and eye blinks appear natural and fluid. ...
2. **Background and Lighting**: The background is a plain curtain, and the lighting is soft and even, consistent with a real indoor setup. ...
3. **Tattoo Detail**: The tattoo on his neck is crisp and well-defined, with natural shading and edges...
<answer>real</answer>

**Gemini-3.0-Pro-Preview** ✅
1. **Inconsistent Skin Texture:** There is a noticeable discrepancy in detail between the frames. In the first frame (00:00), the subject's skin has high-frequency details like pores and individual beard hairs. However, in the frames where the head moves (00:02 and 00:03), the skin texture suddenly becomes overly smooth, waxy looking. ...
2. **Morphing Details (The Tattoo):** The tattoo on the subject's neck reveals a significant artifact. In the earlier frames, it appears to have a distinct script-like structure. In frame 00:03, as the head turns, the tattoo loses its coherence and "melts" into a blurry, dark smudge. ...
<answer>fake</answer>

**BusterX++** ❌
<think> Let's analyze this step by step to determine whether the video is real or fake:
1. **Visual Cues**: The individual in the video appears to have natural skin texture, facial features, and hair that look consistent with a real person. ...
2. **Lighting and Shadows**: The lighting is even and natural, casting soft shadows that are consistent with real-life lighting conditions. ...
3. **Background**: The background appears to be a real-life setting, possibly a room with curtains. ...
4. **Facial Features and Expressions**: The person's facial features, including the eyes, nose, and mouth, show natural movement and expressions that are typical of a real person speaking....
Therefore, the conclusion is that this video is real.
The answer is: A) real </think>
<answer>A</answer>

**Skyra-RL** ❌
<think>I am conducting a thorough analysis of this video to verify its authenticity. The video features a man speaking directly to the camera against a plain, light-colored background. His movements, facial expressions, and speech appear natural. I will now perform a detailed inspection for any potential signs of digital manipulation or AI generation. My focus is on the background, as this can often reveal subtle artifacts. Specifically, I am examining the area of the background behind the speaker <t>[0.0, 6.12]</t> at <bbox>[0.0, 0.0, 1.0, 0.8]</bbox>. Upon close inspection, the background is completely stable and consistent throughout the video. The texture is uniform, and there is no evidence of any unnatural noise, jitter, or structural distortion. The lighting on the background is also stable, with no flickering or inconsistent shadows. The video appears to be a standard, authentic recording. My review finds no artifacts.</think>
<answer>Real</answer>

**VideoVeritas (ours):** ✅
At first glance, the video appears to be a standard, slightly stylized interview or vlog segment. The man's expressions and the overall lighting seem plausible, making it look like a genuine recording at first.
Despite the initial impression of realism, a closer examination reveals significant anomalies, primarily concerning the subject's facial features. The most prominent issue is the behavior of the eyes. Throughout the video, the man's eyes exhibit a subtle but persistent unnatural quality. **For instance, from <time> 1.2s-2.5s </time>, as he looks up, his eyes appear slightly too wide** and **have a glassy, overly smooth look** that lacks the fine texture and natural moisture of real eyes. This is particularly noticeable in the way his eyelids move.
More definitively, there is a clear structural failure in the mouth and teeth area. **When the man opens his mouth to speak, especially between <time> 2.1s-3.3s </time>, the teeth that are visible do not maintain a stable form.** Instead of being solid, well-defined objects, they appear to warp, stretch, and momentarily lose their distinct shape, almost like a digital texture being deformed. This is a classic artifact of a generative model struggling to maintain the complex geometry of a detailed object during motion. ...
But wait, there's an even more subtle flaw in the facial structure. While the mouth is the most obvious area of failure, a similar instability can be observed on the right side of his face. **As his head tilts down around <time> 3.8s-4.5s </time>, the contour of his right cheek and the area around his right eye seems to subtly warp and lose definition**, not just in terms of the mouth, but in the overall geometry of his face. This points to a broader failure in maintaining consistent facial structure, where different parts of the face exhibit similar, localized instability.
The video is definitively identified as AI-generated. While it maintains a superficial level of realism, it is betrayed by multiple, clear signs of digital manipulation. ...
<answer>fake</answer>



**Realistic video, facial**

**Qwen3-VL-8B** ✅
This video appears to be real. The lighting, facial expressions, and slight movements of the person are natural and consistent with a real interview or documentary-style footage. The "FIFA TV" watermark in the corner also suggests it's from an official source, which further supports its authenticity. There are no obvious signs of AI manipulation or deepfake artifacts.
<answer>real</answer>

**Gemini-3.0-Pro-Preview** ✅
1. **High-Fidelity Skin Texture:** The video displays extremely realistic and complex skin details. You can see individual pores, variations in skin tone, wrinkles, and a natural sheen of sweat or oil on the forehead and nose....
2. **Natural Facial Hair:** The stubble on the subject's face is rendered with individual hairs visible, showing realistic density and direction. ...
3. **Realistic Motion and Physics:** The blinking motion (seen in the fourth frame) is physically accurate. ...
<answer>real</answer>

**BusterX++** ✅
<think> Let's analyze this video step by step to determine if it is real or fake:
1. **Visual Quality and Clarity**: The video appears to be a high-quality, professional production. The lighting is well-balanced, highlighting the subject's face without harsh shadows or overexposure. ...
2. **Background and Setting**: The background is blurred, which is a common technique used in professional video productions to keep the focus on the subject. ...
3. **Watermark and Branding**: There is a watermark in the top right corner that reads "FIFA TV." This indicates that the video is associated with FIFA, the international governing body of football. ...
The answer is: A) real </think>
<answer>A</answer>

**Skyra-RL** ✅
<think>I am conducting a thorough analysis of this video, which features a man being interviewed in what appears to be a studio setting. My goal is to meticulously check for any signs of digital manipulation or AI generation. The subject's movements, the camera focus, and the lighting all appear natural. I will now perform a targeted inspection of areas commonly affected by AI generation. Specifically, I am examining the top right corner of the frame, as this is a potential location for overlays in a broadcast video. I am checking the area <t>[0.0, 6.12]</t> at <bbox>[0.7, 0.0, 1.0, 0.3]</bbox>. In this region, I can see the "FIFATV" logo. The text is perfectly clear, stable, and legible throughout the entire clip. It does not flicker, warp, or distort in any way. The presence of this crisp logo in the corner is consistent with a genuine broadcast. Having found no anomalies, I conclude the video is authentic.</think>
<answer>Real</answer>

**VideoVeritas (ours):** ✅
My initial impression is that this is a genuine, professionally shot interview. The subject's expressions, skin texture, and the overall lighting appear natural and consistent with a real-world recording.
The video's authenticity is strongly supported by the complex and subtle biological details present. The subject's facial movements are entirely natural and synchronized with the cadence of speech. For instance, **the way his lips purse and his jaw subtly shifts as he speaks, particularly visible around <time> 0.5s-1.5s </time>, is fluid and biomechanically correct**. His blinking is not uniform; it's a natural, asynchronous process, **with a noticeable blink at <time> 0.8s </time> that is perfectly timed with a slight pause in his speech.**
Furthermore, the physical properties of the scene are rendered with high fidelity. The lighting, which appears to be a soft key light from the front-left, creates consistent highlights and shadows. The specular highlights on his forehead and nose catch the light realistically, and the shadows under his nose and chin deepen and soften in a way that corresponds to the subtle movements of his face. The skin texture itself is a key indicator; it's not a uniform, "painted-on" surface. Instead, **it shows fine pores, natural creases around the eyes**, and a slight sheen that is consistent with the lighting and the subject's skin type. These are complex, stable details that are difficult for generative models to replicate without introducing artifacts.
The video is conclusively real. The evidence lies in the combination of natural, non-repetitive biological motion, including micro-expressions and blinking patterns ...
<answer>real</answer>

*Figure 24.* Reasoning comparisons between **VIDEOVERITAS** and existing MLLM-based detectors on facial videos.

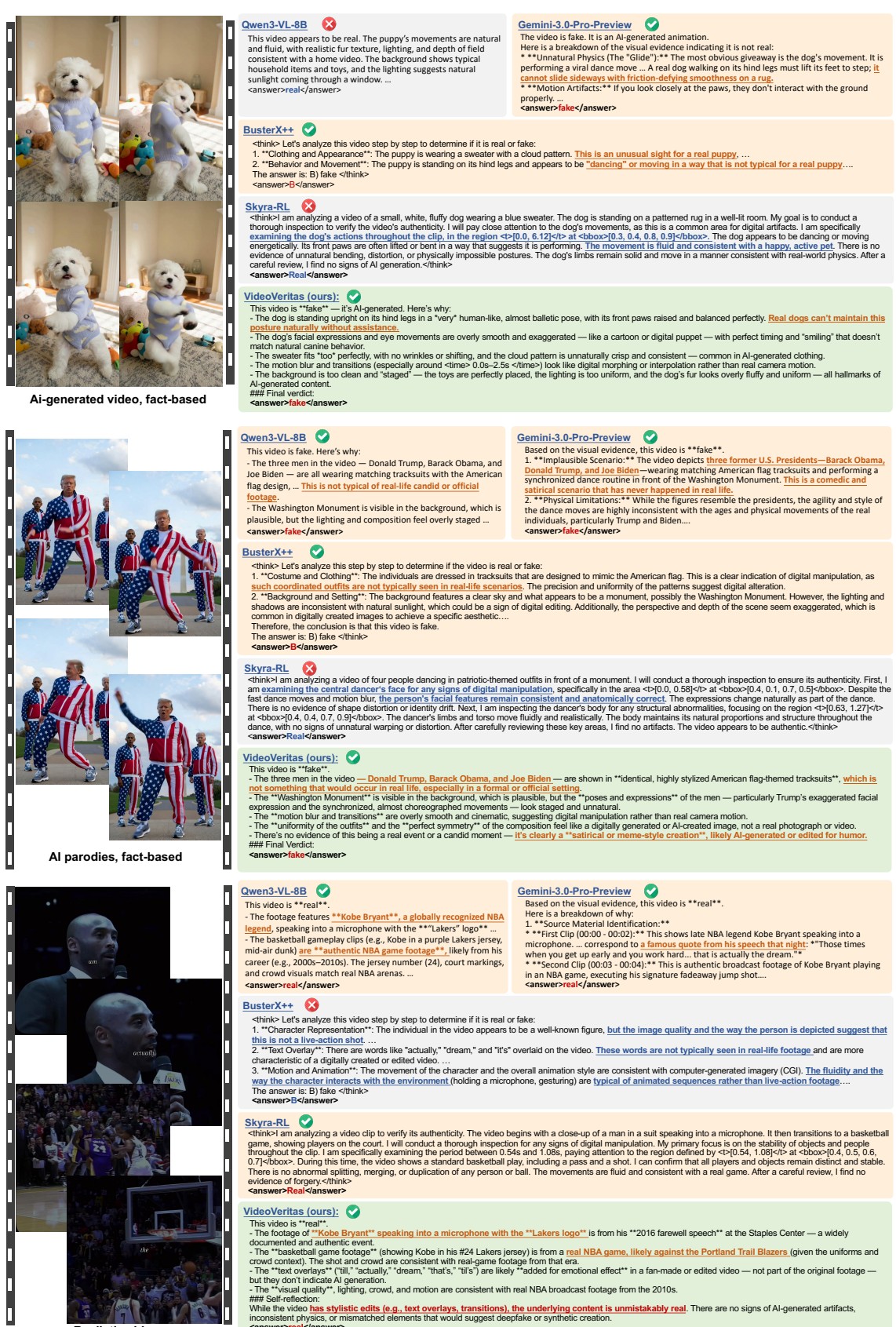

*Figure 25.* Reasoning comparisons between VIDEOVERITAS and existing MLLM-based detectors on fact-based videos.

