# OpenReview forum: "VideoVeritas: AI-Generated Video Detection via Perception Pretext Reinforcement Learning"
_ICML.cc/2026/Conference — ICML 2026 regular_

### Official Review · Reviewer_F5UG · 2026-03-10

**Soundness:** 3
**Presentation:** 2
**Significance:** 3
**Originality:** 3
**Overall Recommendation:** 4
**Confidence:** 3

**Summary:**

This paper addresses the critical security risks posed by the rapid advancement of AI video generation technology, focusing on the core task of AI-generated video detection. To resolve the limited fine-grained perception of current multi-modal large language models (MLLMs), the authors propose VideoVeritas, an end-to-end detection framework with a two-stage training paradigm. In the first stage, the model completes cold start via Joint Preference Alignment (Joint DPO), which integrates both response-level and video-level alignment to balance the model's fine-grained perception and fact-based reasoning capabilities. In the second stage, the authors innovatively propose Perception Pretext Reinforcement Learning (PPRL), which optimizes the model through two general perception pretext tasks (spatiotemporal grounding and self-supervised object counting) instead of directly optimizing for the detection task, thus fundamentally enhancing the model's fine-grained visual perception. Additionally, the paper constructs MintVid, a high-quality evaluation dataset containing 3K videos from 9 state-of-the-art generators, covering three evaluation subsets (general content, facial videos, and factual errors), which fills the gap of outdated generation models and single evaluation dimensions in existing datasets. Extensive experimental results show that VideoVeritas achieves more balanced and superior detection performance on both in-domain and out-of-domain benchmarks, addressing the limitation of existing methods that are biased towards either superficial reasoning or mechanical analysis.

**Compliance With Llm Reviewing Policy:**

Affirmed.

**Key Questions For Authors:**

1.Could you please clarify the full expanded name of the abbreviation "J-DPO" in this work? Does SFT reduce the reasoning ability while J-DPO inject fine-grained perceptual ability and preserve general fact-based reasoning ability at the same time?

2.The paper states that video-level alignment benefits the synergy of fact-based reasoning ability. Could you elaborate on the intrinsic mechanism of how video-level alignment enhances and preserves the model's fact-based reasoning capability, along with the corresponding ablation studies and quantitative results that support this claim? In addition, please explain the respective roles and synergy between response-level alignment and video-level alignment in maintaining and boosting fact-based reasoning.

3.The core of your Joint Preference Alignment module relies on differentiated preference pairing for two types of videos: vp (videos requiring fine-grained perception) and vr (videos requiring fact-based reasoning). Could you detail the way to obtain these two sample sets?

4.The paper mentions that it is non-trivial to directly perform batch-level integration of perception and detection tasks, and Table 5 shows that the vanilla batch-level method underperforms the baseline significantly. Could you explain in detail the underlying reasons for the inferior performance of batch-level integration, and specify the core technical challenges that make direct batch-level joint training non-trivial?

**Limitations:**

yes

**Strengths And Weaknesses:**

Strengths

1.	Soundness: The submission is technically solid with rigorous experimental design and comprehensive validation. The PPRL method is appropriate. The experiments are well-designed. The authors conduct extensive ablation studies to verify the effectiveness of each component of the framework, including the contribution of different training stages, the types of perception pretext tasks, the scale of perception data, and the difficulty of pretext tasks. The authors also perform quantitative analysis of the model's reasoning behavior, and compare with other methods across multiple dimensions, providing sufficient empirical evidence to support the central claims of the paper.

2.	Presentation: The paper has a clear structure, a progressive narrative logic, and standardized and complete charts. The overall narrative is easy to follow. The comparison with related work has clear boundaries, and the method details, implementation parameters, and experimental procedures are disclosed, which is convenient for peers to reproduce and expand the research.

3.	Significance: The authors claim to focus on an important theme of AI-generated video detection, which is critical for mitigating the security risks brought by the rapid development of video generation technology. The proposed method effectively balances the fine-grained perception and fact-based reasoning capabilities of the model, and its detection performance on high-fidelity AI-generated videos significantly outperforms existing state-of-the-art methods, with extremely high industrial application value. Meanwhile, the framework and benchmark provided by this work can be reused by follow-up related research, greatly promoting the development of the field.

4.	Originality: This work is the first to introduce perception pretext reinforcement learning into the field of AI-generated video detection, breaking the inherent paradigm of existing methods that perform fine-tuning or reinforcement learning directly on the detection task, and providing a brand-new research perspective for improving the capability of MLLM-based detection models. The proposed MintVid dataset covers the latest commercial and open-source generation models, and adds a factual error evaluation subset, filling the gap in the field's evaluation benchmarks.

Weaknesses

1.	The authors do not provide systematic theoretical analysis or interpretability evidence to explain the intrinsic working mechanism of PPRL. The paper does not theoretically characterize the transferability between the representations learned from general spatiotemporal grounding/self-supervised object counting and the downstream AI-generated video detection task. There is no analysis of why and how the generic perceptual representations learned from pretext tasks can be effectively transferred to identify subtle generation artifacts, which is the core motivation of PPRL.

2.	The paper does not clearly elaborate the design motivation of Joint Preference Alignment (J-DPO), and the intrinsic link between fact-based reasoning capability preservation and the dual DPO alignment design is not explicitly established, leading to ambiguity in the core logic of this module. On one hand, the paper does not clearly explain the underlying mechanism of why standard Supervised Fine-Tuning (SFT) degrades the native fact-based reasoning ability of the base MLLM, nor does it rigorously demonstrate how the proposed differentiated preference pairing and dual-level (response-level + video-level) alignment design of J-DPO can mitigate this degradation. On the other hand, the paper does not clarify the synergy between the two core goals of J-DPO: injecting fine-grained perceptual ability and preserving general fact-based reasoning ability. It is difficult for readers to understand the respective roles of the two alignment objectives in achieving these two goals, and how the module balances the two capabilities rather than creating conflicts between them.

3.	There are several non-trivial technical errors, citation mistakes, and undefined notations in the paper, which undermine the rigor and readability of the manuscript:

(1) The text states that the tracking reward is the mean Intersection over Union (IoU) over all frames, but Equation (4) is formulated as the sum of IoU across frames, not the mean, resulting in an inconsistency between the formula and its description.

(2) Line 367 of the manuscript refers to the ablation results of perception pretext tasks as Table 4, but the corresponding results are actually presented in Table 2, which is a clear citation error.

(3) In the paragraph describing video-level alignment, the notations yp and yr are used without any prior definition. Based on the context and the logical consistency of the formula, these notations should be the previously defined video-type symbols vp and vr, which is a typo with missing definition.

---

> ### Author Rebuttal · Authors · 2026-03-31
>
> > **Q1: Why the generic perceptual representations can be transferred to identify generation artifacts?**
>
> Thanks for the comment. The essential insight is that AI-generated video detection **fundamentally depends on general spatiotemporal abilities**, such as tracking objects across frames (e.g., for *object permanence, identity consistency*), capturing object relations (e.g., for *interaction physics*), and identifying granular components (e.g., for *anatomical correctness*). For example, "tracking a small object across frames" is directly required to detect abrupt disappearance, a common artifact in generated videos. Figure 2 empirically shows that PPRL improves these capacities effectively.
> Furthermore, this improves rollout quality in detection RL: better perception enables more accurate reasoning traces during exploration. Although PPRL does not directly teach fake detection, it provides perceptual prerequisites for **more effective exploration once coupled with detection learning**.
>
> > **Q2:  Clarify the full name of "J-DPO"; Why does SFT degrade fact-based reasoning, and how does J-DPO mitigate this?**
>
> "J-DPO" stands for Joint Direct Preference Optimization.
>
> Standard SFT learns by **absolute target fitting**, enforcing exact token-level imitation. This can degrade generalization in heterogeneous scenarios, e.g., when perceptual analyses dominate the tokens in training data, SFT could bias towards this pattern, degrading results on fact-based subset (-1.2% in Table 3).
> In contrast, J-DPO mitigates this by learning **conditional pairwise preferences** rather than absolute fitting. The model is encouraged to learn the **comparative relationship** across reasoning traces. Such signals are implicitly **more informative** than rigid token-fitting, e.g., the model is encouraged to distinguish precise perception ($y_e$) from hallucinated traces ($y_b$) rather than directly fitting $y_e$, and vice versa. As a result, J-DPO improves perception without erasing factual reasoning, improving both subsets in Table 3.
>
> > **Q3: How does video-level alignment preserve fact-based reasoning? What is the synergy between response-level and video-level alignment?**
>
> In fact, the synergy is not solely established by video-level alignment. As mentioned in **Q2** above, the *comparative preference learning* objective in DPO is the key mechanism to synergize both capacities.
>
> To be specific, $DPO_t$ teaches model "**what better reasoning looks like**", and $DPO_v$ tells model "**when this reasoning should be activated**":
> - Response-level ($DPO_t$): Given a visual condition $v$, it optimizes preference between responses ($\pi_\theta(y_w|v) > \pi_\theta(y_l|v)$). This functions as a capability injector. It teaches model the quality boundaries of specific reasoning path (e.g., distinguishing precise perception from hallucination, producing fact reasoning instead of redundant analysis).
> - Video-level ($DPO_v$): Given a textual reasoning $y$, it further **sharpens** preference across visual conditions ($\pi_\theta(y|v_w) > \pi_\theta(y|v_l)$). This acts as a cross-modal anchor, encouraging model to learn intrinsic dependency between visual cues and preferred reasoning paradigms.
>
> Therefore, in Table 3, $DPO_t$ alone achieves gains on both subsets over SFT. $DPO_v$ further sharpens this behavior through stronger cross-modal anchoring (+1.7% on fact subset over $DPO_t$).
>
> > **Q4: Technical errors and undefined notations.**
>
> Thanks for carefully identifying these issues. We have verified:
> - Eq. (4): We checked our implementation and it should take the **mean**. We will correct Eq. (4) in the revision.
> - L367: It should reference **Table 2**. We will fix this.
> - $y_p$ and $y_r$: These should be $v_p$ and $v_r$. We will correct the notations.
>
> We will proofread the paper to avoid any typos.
>
> > **Q5: How are $v_p$ and $v_r$ obtained?**
> - $v_p$ (perception-type videos): sampled from GenBuster-200K, RewardData, TalkingheadBench and OpenVid.
> - $v_r$ (reasoning-type videos): manually filtered from YouTube, containing both real and fake samples verifiable via objective facts or commonsense.
>
> We will clarify the data sources in the revision.
>
> > **Q6: Why does vanilla batch-level integration of perception and detection underperform?**
>
> This stems from gradient imbalance in GSPO-style objective. The gradient term is $\nabla_\theta L_{GSPO}=-E\left[\frac{1}{G}\sum_{i=1}^G\frac{1}{|o_i|}\sum_{t=1}^{|o_i|}\omega_{i}A_{i,t}r_{i}(\theta)\nabla_\theta\log\pi_\theta(o_{i,t}\mid\cdot)\right]$ ($\omega_{i}$ is clipping indicator).
> The contribution of each token is scaled inversely by output length $\frac{1}{|o_i|}$. In our case, perception tasks output only a few tokens, while detection generates much longer outputs (e.g., 1 vs. 100). Therefore, tokens from perception tasks dominate gradient updates within a mini-batch, leading to inferior performance than training exclusively on detection.
> We will add a gradient-norm analysis to illustrate this.

---

> > ### Author Rebuttal · Reviewer_F5UG · 2026-04-04
> >
> > I feel the additional explanations the authors provided are convincing and I do think this is an interesting paper so I will raise my score to accept.

---

### Official Review · Reviewer_PC54 · 2026-03-10

**Soundness:** 3
**Presentation:** 2
**Significance:** 2
**Originality:** 3
**Overall Recommendation:** 4
**Confidence:** 5

**Summary:**

This paper argues that existing AIGC video detection methods may produce reasoning that does not necessarily align with factual evidence, and that current models often fail to capture artifacts that are perceptible to humans. To address this issue, the authors propose Video Veritas, which first applies DPO to train the model and then further improves performance through a two-stage reinforcement learning framework consisting of Perception RL and Detection RL. Notably, the authors introduce a Perception Pretext RL stage, where the model learns proxy tasks unrelated to AIGC video detection in order to enhance its ability to perceive artifacts visible to humans. Video Veritas significantly improves the model’s detection performance. In addition, the authors introduce a new evaluation benchmark, MintVid, which provides the community with a richer evaluation resource. However, the paper still presents significant hallucination risks in the construction of its training data, particularly in the CoT annotations, and the authors do not provide any measurement or analysis of this issue. Based on these considerations, I consider the paper to fall between **Weak Reject** and **Weak Accept**.

**Compliance With Llm Reviewing Policy:**

Affirmed.

**Final Justification:**

I have no other questions and keep my rating.

**Key Questions For Authors:**

See the content in Weakness.

**Limitations:**

yes

**Strengths And Weaknesses:**

**Strength**
1. The authors propose Video Veritas, a method that enhances AIGC-generated video detection by introducing visual proxy tasks. Specifically, the authors leverage counting and temporal object localization tasks to improve the model’s ability to perceive potential artifacts. Considering that AI-generated videos often exhibit temporal inconsistencies such as flickering or abrupt transitions, it is reasonable that such proxy tasks could improve detection performance.

2. The authors introduce a new evaluation dataset, MintVid, which includes videos generated by more recent generation methods. This dataset is likely to facilitate further research and development within the community.

3. The paper reports good empirical performance and includes a set of ablation studies that provide insights into the contributions of different components of the framework.


**Weakness**
1. Poor description of the training data. The paper lacks a clear and centralized description of the training data and instead scatters relevant information across multiple subsections, making it difficult to understand. A similar issue appears in the Ablation Study on the training stages, where the total training data is stated to be 19K samples, yet the Pure RL setting uses 20K samples, leading to confusion about the data configuration.

2. The authors argue that directly annotating the authenticity of videos is difficult and may lead to mechanical explanations. However, the paper still relies on Qwen-32B to annotate semantic anomalies in the images. This raises my concerns about potential hallucinations. Based on my experience, if a model can reliably identify anomalies, this is almost equivalent to the model being able to determine the authenticity of the data. For example, when anomalies in AIGC data are provided as contextual input to a model, detection accuracy can increase substantially. This phenomenon suggests that the ability to perceive semantic artifacts is nearly equivalent to the ability to perform detection. Therefore, the authors should verify the accuracy and reliability of these CoT annotations.

3. Although the proposed method is demonstrated on videos, the authors should discuss the possibility of transferring the approach to image-level detection. (This is a minor concern.)

4. The proposed framework appears to primarily enhance the model’s ability to detect semantic-level artifacts. However, invisible generator artifacts are also an important signal for detection. Video Veritas does not seem capable of leveraging or enhancing such features. The authors may want to clarify whether this understanding is correct.

---

> ### Author Rebuttal · Authors · 2026-03-31
>
> > **Q1: Poor description of training data; confusion between 19K and 20K samples.**
>
> We apologize for the confusion and thank the reviewer for pointing this out. In the paper, we rounded the data counts for brevity (e.g., "4K" and "10K"), which made the configurations look inconsistent. In the actual setup:
> - J-DPO stage: **4,219** samples, from GenBuster-200K, RewardData, TalkingheadBench, OpenVid, and fact-based videos from YouTube.
> - RL stage: (1) **10,789** AIGC samples, from the same sources of J-DPO, and (2) **5,000** perception samples, where grounding data are sampled from OneThinker, and counting data are constructed from OpenVid.
>
> Therefore, VideoVeritas adopts a total of **20,008** training samples. To keep consistent training scale, Pure RL setting also adopts 20K samples. We will add a **centralized summary of data scales and sources** in the paper to eliminate this ambiguity.
>
> > **Q2: The authors should verify the accuracy and reliability of these CoT annotations.**
>
> Thanks for raising this concern. We agree that the ability to detect semantic anomalies is closely related to forgery detection. However, we observe that the task framing significantly affects how well models identify anomalies. We investigated on AbHuman [1], which provides GT labels for structural anomalies. When Qwen3-VL-32B is prompted to judge the authenticity, the model tends to produce holistic judgments without attending to specific anomalies present (only 64.3% accuracy on AbHuman). In contrast, when the same model is directly asked targeted questions (e.g., *"... Is the right leg distorted?"*), it achieves improved accuracy in anomalies identification (81.9% on AbHuman). A similar finding also motivates VBench-2.0 [2], which adopts decomposed VQA rather than direct judgment for evaluating video generation quality.
>
> Our annotation pipeline is designed precisely around this insight. Rather than asking the model to produce a real/fake judgment, we **decouple** the annotation into anomalies identification and CoT generation. During first stage, each question targets a localized observation, which probes more accurate anomalies identification.
>
> To quantitatively verify the quality, we randomly sampled **500 questions** for **human judgments**. Human annotators were asked to independently answer each question. Results show that the **agreement** between the model's answers and human judgments is **89.8%**, confirming that our pipeline produces reliable QA reports for data curation.
> We will include more details on the quality analyses in the paper.
>
> > **Q3: Transferability to image-level detection.**
>
> Thanks for the suggestion. Although VideoVeritas is designed for video-level detection, its underlying principles, i.e., fine-grained perception and fact-based reasoning, are also applicable to images. We compare the performance on Chameleon [3] and GenBuster++ (image part) [4]:
> |Method|Training|GenBuster++(Acc/Recall/F1)|Chameleon(Acc/Recall/F1)|
> |---|---|---|---|
> |Qwen3-VL-8B|-|62.1/28.9/43.4|72.0/57.2/63.7|
> |FakeVLM [5]|Image|55.5/**86.5**/66.0|62.9/**99.7**/69.7|
> |BusterX++ [4]|Image+Video|78.1/75.7/77.6|76.1/80.5/74.3|
> |**VideoVeritas**|Video|**86.4**/76.2/**84.9**|**79.8**/88.9/**79.1**|
>
> Despite being trained exclusively on video data, VideoVeritas achieves competitive performance on image benchmarks.
>
> > **Q4: VideoVeritas does not seem capable of leveraging invisible generator artifacts, the authors should clarify whether this understanding is correct.**
>
> We sincerely thank the reviewer for this insightful question. We agree that VideoVeritas is more effective at capturing perceptible spatiotemporal artifacts. If future generators could produce "completely flawless" videos, then current detectors would indeed face challenges. However, as our experiments in Table 4 suggest, machines have the potential to surpass the physiological limit of human perception (Sec 4.4, L380: "Hard" setting is already difficult for humans to perceive, yet the model can still be trained to count them accurately). This indicates that, with appropriate task design, MLLMs still have the potential to be trained to capture pixel-level digital artifacts, which are human-imperceptible.
>
> Besides perception, a key strength of VideoVeritas is fact-based reasoning, which is especially important for videos that are perceptually plausible in appearance but violate objective facts or commonsense. We will clarify this scope more explicitly in the paper.
>
> > **References**
>
> [1] Humanrefiner: Benchmarking abnormal human generation and refining with coarse-to-fine pose-reversible guidance. ECCV 2024
>
> [2] VBench-2.0: Advancing video generation benchmark suite for intrinsic faithfulness.
>
> [3] A Sanity Check for AI-generated Image Detection. ICLR 2025
>
> [4] BusterX++: Towards Unified Cross-Modal AI-Generated Content Detection and Explanation with MLLM.
>
> [5] Spot the Fake: Large Multimodal Model-Based Synthetic Image Detection with Artifact Explanation. NeurIPS 2025

---

> > ### Author Rebuttal · Reviewer_PC54 · 2026-04-03
> >
> > I have no other questions and keep my rating.

---

### Official Review · Reviewer_xjs4 · 2026-03-12

**Soundness:** 3
**Presentation:** 4
**Significance:** 3
**Originality:** 3
**Overall Recommendation:** 5
**Confidence:** 4

**Summary:**

This paper focuses on detection of AI generated videos. It introduces a method for detection of fake videos which not only detects them but can also give a good explanation of why the system decides that it has been AI generated. The focus of the method is on fine-grained details in the video. A relatively small but solid new dataset for validation is proposed.

**Compliance With Llm Reviewing Policy:**

Affirmed.

**Final Justification:**

I found the paper already interesting and a good contribution. The rebuttal and the new experiments done will strengthen the final paper. Hence this paper is an accept for me.

**Key Questions For Authors:**

1. For me the most interesting and convincing part is the explanations that the method can generate. There is only one example in the main text body. Can't you define a metric to assess the quality of the descriptions, maybe including some human judgments on a limited set of examples?
2. What are examples of failure cases? Are there any hallucinations in the qualitative descriptions? Failure cases should consider both wrong detections as well as correct detections for the wrong reasons.

**Limitations:**

See key question 2. Focus is on successful results.

**Strengths And Weaknesses:**

Strengths:
- solid reinforcement learning based methodology focusing on fine-grained details in the video. Good use of pretext training tasks vs deployment.
- Good performance on in-domain and out-of-domain testing as well as out-of-domain testing on their own proposed dataset.
- Convincing qualitative study showing superior explanations over existing methods.

Weaknesses:
- Limited qualitative examples in the main paper
- No examples of failure cases

---

> ### Author Rebuttal · Authors · 2026-03-31
>
> > **Q1: Limited qualitative examples in the main paper; include some human judgments on a limited set of examples.**
>
> Thanks for this suggestion. Due to page limits, we included only one qualitative example in the main paper. In the Appendix, we provide extensive qualitative examples covering high-quality and low-quality generated videos, deepfake videos, AI-parody videos, and corresponding real videos, demonstrating VideoVeritas's strong perception and analysis capabilities across object deformation, disappearance, and contextual inconsistencies, etc.
>
> To quantitatively evaluate explanation quality, we conducted both **MLLM-as-a-Judge** and **Human Judgment** on VideoVeritas, Skyra, and BusterX++. Following AIGI-Holmes [1], the final scores are computed through Elo rating.
>
> - For MLLM-as-a-Judge, we randomly sampled 1,000 videos and their reasoning outputs (800 fake, 200 real) and utilized Gemini-3.0-Pro-Preview for judgment.
> - For human judgments, we randomly sampled 200 videos (150 fake, 50 real) for pairwise comparisons (resulting in 600 pairwise samples for three models). The human annotators were asked to decide which explanation is better given outputs from two different models. Due to time limits, we did not involve complex rubrics but ask the annotators to decide through **three perspectives**: *(1) perception accuracy* (e.g., whether the mentioned anomalies actually exist), *(2) forensic objectivity* (e.g., prefer objective descriptions than subjective feelings), and *(3) logical coherence* (e.g., are there any contradictions in the reasoning).
>
> | Method | MLLM-as-a-Judge | Human Judgment |
> |---|---|---|
> | Skyra | 784.30 | 724.50 |
> | BusterX++ | 963.29 | 1089.21 |
> | **VideoVeritas** | 1252.41 | 1186.30 |
>
> The results show that **VideoVeritas outperforms existing detectors in reasoning quality** under both evaluation protocols, which align with our qualitative observations. We will report more details of MLLM-as-a-Judge (e.g., full input prompt) and Human Judgment (e.g., annotation platform) in the revised paper.
>
> > **Q2: No examples of failure cases. Failure cases should consider both wrong detections as well as correct detections for the wrong reasons.**
>
> Thanks for the suggestion. We analyzed the model's failure cases and identified three representative categories. Due to the format constraints, we provide a summary of the key failure modes here. Complete examples and a thorough discussion of failure modes will be included in our revised paper.
>
> **(1) False negatives.** These occur when the generated video is of near-perfect visual quality. However, we note that in these cases, the model **still raises specific concerns** in its reasoning chain (e.g., "... make it an authentic video, but the gelatinous texture of the water splashes *still raises a red flag*. Overall, this is a hard case ..."), which can serve as useful flags for further scrutiny or future improvements.
>
> **(2) False positives.** Real videos can be misclassified as fake mainly in two scenarios: (a) The quality of the video is poor, containing unexpected noise such as heavy blur. (b) Videos from short-video platforms that contain special effects or dense text overlays. For example, some users may intentionally add a Sora watermark onto a real video. Although these videos are not AI-generated, they still undergo severe post-processing, which necessitates a clearer boundary definition.
>
> **(3) Correct label with partially inaccurate reasoning.** The model may correctly predict the video's authenticity while the reasoning is partially inaccurate. We found that this mainly occurs on low-level descriptions such as texture and surface stability ("... this graphic exhibits a subtle but persistent 'boiling' or shimmering effect, as if the pixels are not stable ..."). On one hand, these **pixel-level instabilities** may indeed remain undetected by human eye, making it a potentially valid indicator. On the other hand, the final judgment is consistently determined by synthesizing and weighing different evidence, showing robustness when encountering ambiguous cases.
>
> We will add representative examples and discussions in the revised paper.
>
> > **References**
>
> [1] AIGI-Holmes: Multi-modal AI-Generated Image Detection with Explanatory Reasoning. ICCV 2025

---

> > ### Author Rebuttal · Reviewer_xjs4 · 2026-04-02
> >
> > I feel the additional experiments the authors conducted to answer are convincing and I do think this is an interesting paper so I will raise my score to accept.

---

### Official Review · Reviewer_DnXJ · 2026-03-12

**Soundness:** 3
**Presentation:** 3
**Significance:** 3
**Originality:** 3
**Overall Recommendation:** 4
**Confidence:** 5

**Summary:**

This paper builds its framework around what I believe are the two fundamental abilities required for fake video detection: fine-grained perceptual sensitivity and logical/factual reasoning. This overall motivation is intuitive and consistent with how humans would think about the problem. In addition, the paper introduces a more challenging and practically meaningful benchmark, which is valuable. However, I still have several concerns about whether the key design choices are fully justified. In particular, I am not convinced by the way the DPO stage separates perception and reasoning, by the explanation of the PPRL design, and by the potential bias introduced by teacher-generated QA reports and CoT supervision.

**Compliance With Llm Reviewing Policy:**

Affirmed.

**Final Justification:**

The rebuttal addressed my concerns, my final recommendation is "weak accept".

**Key Questions For Authors:**

see weaknesses

**Limitations:**

Yes

**Strengths And Weaknesses:**

Strengths:
1. The paper designs the whole framework starting from the core abilities required for fake video detection, namely fine-grained perceptual sensitivity and logical/factual reasoning. This is consistent with human intuition.
2. The paper proposes a more difficult and practically meaningful benchmark.

Weaknesses:
1. In the DPO stage, the paper seems to decompose the problem into either fine-grained perception or logical/factual reasoning. But what if a video requires both abilities at the same time? For example, in a video, the face of a person in a certain segment may be manipulated, such as expression editing or face swapping. In such a case, correct detection may require both fine-grained perceptual sensitivity and logical/factual reasoning. From the design perspective, the paper does not seem to consider this type of case, which makes its generalization ability questionable.

2. The General training part of the proposed PPRL uses general-domain data and trains general fine-grained visual perception ability. However, strong MLLMs such as GPT-5 and Qwen3-VL-235B have also gone through large-scale fine-grained visual perception training, yet their performance in fake video detection is still poor. In that case, it is difficult to explain why only a few thousand samples in PPRL can substantially improve the model’s fine-grained visual perception ability. I believe the fine-grained visual training received by these large MLLMs must be much stronger and more sufficient than what is used in PPRL.

3. Part of the data used in DPO training is generated by models such as Gemini 2.5, and this is done with prompts based on QA reports. This may introduce a bias of “finding reasons for a known answer” into the final CoT. Then how is this bias handled, and how is its influence minimized? The paper does not address this issue.

---

> ### Author Rebuttal · Authors · 2026-03-31
>
> > **Q1: What if a video requires both fine-grained perception and logical/factual reasoning simultaneously?**
>
> Thanks for raising this concern. In practice, VideoVeritas is capable of performing fine-grained analysis and fact-based reasoning **simultaneously** within a single inference. As shown in **Figure 24**, when analyzing the "dancing dog" video, VideoVeritas simultaneously:
>
> - Reasons about factual plausibility: "Real dogs can't maintain this posture naturally without assistance"
> - Analyzes fine-grained visual details: "The dog's facial expressions and eye movements" and "The motion blur and transitions (especially around 0.0s–2.5s) look like digital morphing or interpolation rather than real camera motion".
>
> The final verdict is based on both kinds of evidence. Similar examples also appear throughout Figure 24.
>
> The core issue here is that existing methods fail to capture such joint capacities, e.g., Skyra cannot reason about implausible real-world behaviors (e.g., unnatural dog postures), while BusterX++ cannot zoom in on fine-grained visual details (e.g., facial expressions and eye movements). VideoVeritas is trained to adopt both capabilities and to deploy them jointly during inference, thereby achieving better generalization.
>
> > **Q2: The fine-grained visual training received by strong MLLMs must be much stronger and more sufficient than what is used in PPRL.**
>
> Thanks for the insightful comment. We agree that strong proprietary MLLMs (e.g., Gemini) indeed exhibit stronger fine-grained perception capacity. As shown in table below, we observe that Gemini-3.0-Pro-Preview shows a clear advantage across perception tasks and achieves a significant lead in detection performance (e.g., +16.0% over Qwen3-VL-235B-A22B on MintVid).
>
> | Model | Counting (Medium) | Counting (Hard) | Tracking | Acc. on MintVid |
> |---|---|---|---|---|
> | Gemini-3.0-Pro-Preview | 79.4 | 50.8 | 42.4 | 72.1 |
> | Qwen3-VL-235B-A22B | 79.5 | 25.5 | 36.5 | 56.1 |
> | GLM-4.5V (106B, A12B) | 75.1 | 34.7 | 39.8 | 56.6 |
> | **VideoVeritas (8B)** | 93.5 | 72.9 | 48.3 | 80.9 |
>
> However, (1) even Gemini-3.0-Pro-Preview struggles on challenging perception settings (e.g., **50.8** on Counting-Hard), and (2) open-source models (e.g., Qwen3-VL-235B-A22B) still significantly lag behind those proprietary models, almost fail on perception-heavy tasks (e.g., only **25.5** on Counting-Hard). These results indicate that large-scale pretraining **still falls short in modeling granular spatiotemporal perception**, particularly for subtle and transient targets. A similar finding is also delivered in a recent study [1].
>
> Unlike generic pretraining, PPRL explicitly focuses on *hard, localized, and temporally precise perception signals* (e.g., short-duration object counting and fine-grained grounding). These signals are underrepresented in standard web-scale data but are highly aligned with detection needs. Therefore, even a few thousand samples can provide strong complementary supervision, yielding substantial gains on perception-heavy tasks (e.g., **72.9** on Counting-Hard) and improved detection performance (+8.8% on MintVid). We will add a discussion in the paper.
>
> > **Q3: DPO training data generated by Gemini may introduce "finding reasons for a known answer" bias.**
>
> Thanks for the comment. We mitigate this issue from two progressive perspectives:
>
> Firstly, the QA stage is built from localized questions that do not presuppose the final real/fake label, e.g., "When the vendor's hand touches the sandwich or bread, does it pass through or merge with the object?". This decouples evidence collection from label prediction and reduces label-confirming hallucinations.
>
> Secondly, to further avoid "reasoning to justify an answer", we explicitly give instructions in the prompt of Gemini-2.5-Pro:
>
> > > "... Note that while the Q&A Report gives you prior information, you must NOT reference or rely on them directly in your explanation. Instead, use them to **silently** guide your reasoning simulation. These should **appear as if made naturally based on the given video alone**. ..."
>
> This enforces that the final CoT must appear as **naturally grounded in the video itself**, rather than derived from external hints.
> We will include the full prompt in the paper to clarify this design.
>
> > **References**
>
> [1] Molmo2: Open Weights and Data for Vision-Language Models with Video Understanding and Grounding.

---

> > ### Author Rebuttal · Reviewer_DnXJ · 2026-04-03
> >
> > Thank the authors' reponse, my concerns are addressed. I will keep my score.

---

### Decision · Program_Chairs · 2026-04-30

**Decision:**

Accept (regular)

**Comment:**

This paper introduces VideoVeritas, an approach for AI-generated video detection using perception pretext tasks with RL. The approach was appreciated for tackling a timely and important problem, a novel perception pretext RL training signal, strong empirical results, and a principled J-DPO formulation that captures response-level and video-level alignment. Some of the raised concerns were: generalizability to unseen generation models, evaluation primarily on current generation methods, batch-level integration limitations need gradient-norm analysis in final paper. The authors did a good job in addressing most of the concerns in their rebuttal, and consequently, two reviewers upgraded their scores. Technical issues are minor and mostly addressed. Reviewers have a consensus to accept the work, and AC agrees.